# Nuclear S-nitrosylation impacts tissue regeneration in zebrafish

Gianfranco Matrone[1,2 ✉], Sung Yun Jung [3,4], Jong Min Choi[3], Antrix Jain [3], Hon-Chiu Eastwood Leung[5], Kimal Rajapakshe[5], Cristian Coarfa [5], Julie Rodor[1], Martin A. Denvir[1], Andrew H. Baker [1] & John P. Cooke [2]

Despite the importance of nitric oxide signaling in multiple biological processes, its role in tissue regeneration remains largely unexplored. Here, we provide evidence that inducible nitric oxide synthase (iNos) translocates to the nucleus during zebrafish tailfin regeneration and is associated with alterations in the nuclear S-nitrosylated proteome. iNos inhibitors or nitric oxide scavengers reduce protein S-nitrosylation and impair tailfin regeneration. Liquid chromatography/tandem mass spectrometry reveals an increase of up to 11-fold in the number of S-nitrosylated proteins during regeneration. Among these, Kdm1a, a well-known epigenetic modifier, is S-nitrosylated on Cys334. This alters Kdm1a binding to the CoRest complex, thus impairing its H3K4 demethylase activity, which is a response specific to the endothelial compartment. Rescue experiments show S-nitrosylation is essential for tailfin regeneration, and we identify downstream endothelial targets of Kdm1a S-nitrosylation. In this work, we define S-nitrosylation as an essential post-translational modification in tissue regeneration.

[1] British Heart Foundation Centre for Cardiovascular Science, Queen's Medical Research Institute, The University of Edinburgh, 47 Little France Crescent, Edinburgh EH16 4TJ, UK. [2] Center for Cardiovascular Regeneration, Department of Cardiovascular Sciences, Houston Methodist Research Institute, Houston, TX 77030, USA. [3] Mass Spectrometry Proteomics Core, Baylor College of Medicine, Houston, TX 77030, USA. [4] Department of Biochemistry and Molecular Biology, Baylor College of Medicine, Houston, TX 77030, USA. [5] Department of Molecular & Cellular Biology, Baylor College of Medicine, Houston, TX 77030, USA. ✉email: gianfranco.matrone@ed.ac.uk

Complete regrowth of functional tissue is a highly desirable, but mostly unachieved, therapeutic target in the tissue loss associated with many human diseases. Innate immune activation is an early response to tissue stress and injury[1]. In lower vertebrates, this is typically followed by functional tissue regeneration while in higher vertebrates there is normally fibrous scar formation[2]. Although several molecular pathways have been implicated in tissue regeneration, the mechanisms underlying this process are still not clearly understood[3].

The interaction between pattern recognition receptors (PRRs) and damage-associated molecular patterns (DAMPs) during and after an injury activates molecular pathways and transcriptional factors that regulate the expression of a plethora of genes. We have previously shown that this inflammatory signaling causes global changes in the expression and post-translational modifications (PTM) of epigenetic modifiers favoring an open chromatin configuration and cellular plasticity[4,5]. It was also found that inducible nitric oxide synthase (iNos) translocates to the nucleus to bind and S-nitrosylate the polycomb and NuRD complexes during trans-differentiation of fibroblasts into endothelial cells[6]. The effect of cell autonomous innate immune signaling to increase DNA accessibility and thereby to facilitate nuclear reprogramming of cell fate is termed transflammation[7]. The role of this phenomenon in tissue regeneration remains unexplored.

Accordingly, here we investigated the role of the innate immune effector iNos and S-nitrosylation of nuclear proteins in zebrafish tailfin regeneration, an ideal model to study appendage regeneration[8]. We found that *inos* translocates from the cytoplasm to the nucleus in the regenerating tailfin and this is associated with an increase in S-nitrosylation of over 500 different nuclear proteins. Of these, we demonstrated a strong link between Kdm1a S-nitrosylation and histone demethylase activity during tailfin regeneration, specifically in endothelial cells, the main target where S-nitrosylation mostly occurs and where the S-nitrosylated form of Kdm1a promotes the expression of proangiogenic genes. Here, we show the essential role of the S-nitrosylation of nuclear proteins in tissue regeneration.

## Results

### Nos2 translocates from the cytoplasm to the nucleus and triggers protein S-nitrosylation during tailfin regeneration.

Compared to mammals that have three *nos* genes (neuronal, or *nos1*; inducible, or *nos2*; and endothelial, or *nos3*), the zebrafish genome has only *nos1* and two *nos2* genes (*nos2a* and *nos2b*). Activation of the innate immune system transcription factor Nf-kb following injury triggers the translocation of the kB subunit into the nucleus which in turn promotes activation of a panoply of genes, including *inos*[9]. The activation of Nf-kb after tailfin amputation was measured as GFP signal in the *Tg(nfkb:EGFP)nc1* zebrafish (Supplementary Fig. S1A, B). Real time PCR for *nos* genes measured in the adult zebrafish (*Danio rerio*) tailfin uninjured (baseline) and at 3, 5, and 10 days of post-amputation (dpa) revealed a significant increase of *nos2b* expression and a slight increase in *nos1* at 3 and 5 dpa (Fig. 1A). Increased *nos2* expression during regeneration was confirmed by western blotting (WB) (Fig. 1B). To confirm the role of Nf-kb on the upregulation of *nos2*, adult zebrafish were injected with the Nf-kb inhibitor Bay11-7082 30 mM, which resulted in a reduced expression of *nos2b* in the injured tailfin (Supplementary Fig. S1C) at 3 and 5 dpa compared to injured control. While iNos has been considered predominantly cytosolic, it has also been localized to other cellular compartments, including the nucleus[6]. Analysis of Nos2 compartmentalization in the uninjured tailfin

confirmed a predominantly cytoplasmic distribution of this protein (Fig. 1C). However, at 3 dpa Nos2 was predominantly in the nucleus, with equal distribution between nucleus and cytoplasm at 5 dpa, returning predominantly to the cytoplasm by 10 dpa, similar to uninjured controls. Thus, immediately after injury, Nos2 translocates to the nucleus and, during later stages of repair, shifts back to the cytoplasm. This key series of observations gave rise to our principal hypothesis, namely, that changes in the distribution of Nos2 during regeneration mirror and drive changes in S-nitrosylation of nuclear proteins.

To address this, we collected newly formed tissue at the wound edge for extraction of nuclear protein after amputation of adult zebrafish tailfins. Separation of nuclear proteins was confirmed by western blotting to detect nuclear and cytoplasmic markers (Supplementary Fig. S1D). The nuclear fractions were treated with iodoacetyl Tandem Mass Tags (iodoTMT) to label S-nitrosylated proteins, and were subsequently identified using an anti-TMT antibody. Western blotting revealed an increase in the number and intensity of protein bands in the regenerating tissue compared to control tissue, suggesting an S-nitrosylation switch in nuclear proteins (Fig. 1D). When zebrafish were injected in the retro-orbital vein with increasing concentrations of the Nos inhibitor N(ω)-nitro-L-arginine methyl ester (L-NAME) or the NO scavenger 2-Phenyl-4,4,5,5-tetramethylimidazoline-1-oxyl 3-oxide (PTIO), the levels of protein S-nitrosylation were reduced (Fig. 1E), whereas treatment with increasing concentrations of the NO donor S-Nitroso-N-acetyl-DL-penicillamine (SNAP) increased protein S-nitrosylation (Fig. 1E). L-NAME and PTIO significantly reduced tailfin regeneration compared to control or SNAP treated groups at 3 and 7 dpa (Fig. 1F and Supplementary Fig. S2A). As L-NAME inhibits all Nos isoenzymes, in order to assess the specific role of iNos we treated the tailfin regeneration model with the iNos selective inhibitor 1400W[10]. Treatment with 1400W 50 mM reduced tailfin regeneration in a manner similar to L-NAME, suggesting that Nos2 is the main Nos isoenzyme involved in the effect observed by L-NAME (Supplementary Fig. S2B). Overall, these results indicate that Nos2 and NO are necessary for tailfin regeneration and that protein S-nitrosylation plays a key role in this process.

### Analysis and screening of S-nitroso-proteome revealed an increase in S-nitrosylated nuclear proteins during tailfin regeneration.

To identify nuclear proteins that were S-nitrosylated during regeneration, we performed liquid chromatography and tandem mass-spectrometry (LC/MS/MS) (Fig. 2A) on protein extracts from the tailfin wound edge, excised at 3, 5, and 10 dpa. Hierarchical clustering showed a striking increase in the number of nuclear S-nitrosylated proteins during regeneration, in particular at 5 dpa (Fig. 2B and Supplementary Fig. S3). We found that the number of nuclear S-nitrosylated proteins increased from 31 in uninjured to 199, 351, and 264 at 3, 5, and 10 dpa, respectively (Fig. 2C). Taking account of the fact that some proteins changed their S-nitrosylation state on more than one cysteine residue of that protein, the actual total number of S-nitrosylated peptides increased from 31 in uninjured to 332, 566, and 450 at 3, 5, and 10 dpa, respectively (Supplementary Fig. S3A). While a few proteins were S-nitrosylated throughout these time-points, suggesting constitutive S-nitrosylation, the analysis revealed that the majority of the proteins were modified uniquely at a specific time-point. These data strongly suggest a dynamic choreography of S-nitrosylation throughout the regeneration process. Indeed, an analysis of differential enrichment revealed that proteins of many different pathways are S-nitrosylated specifically during regeneration (Fig. 2D). Some of these are known to be implicated in developmental and wound

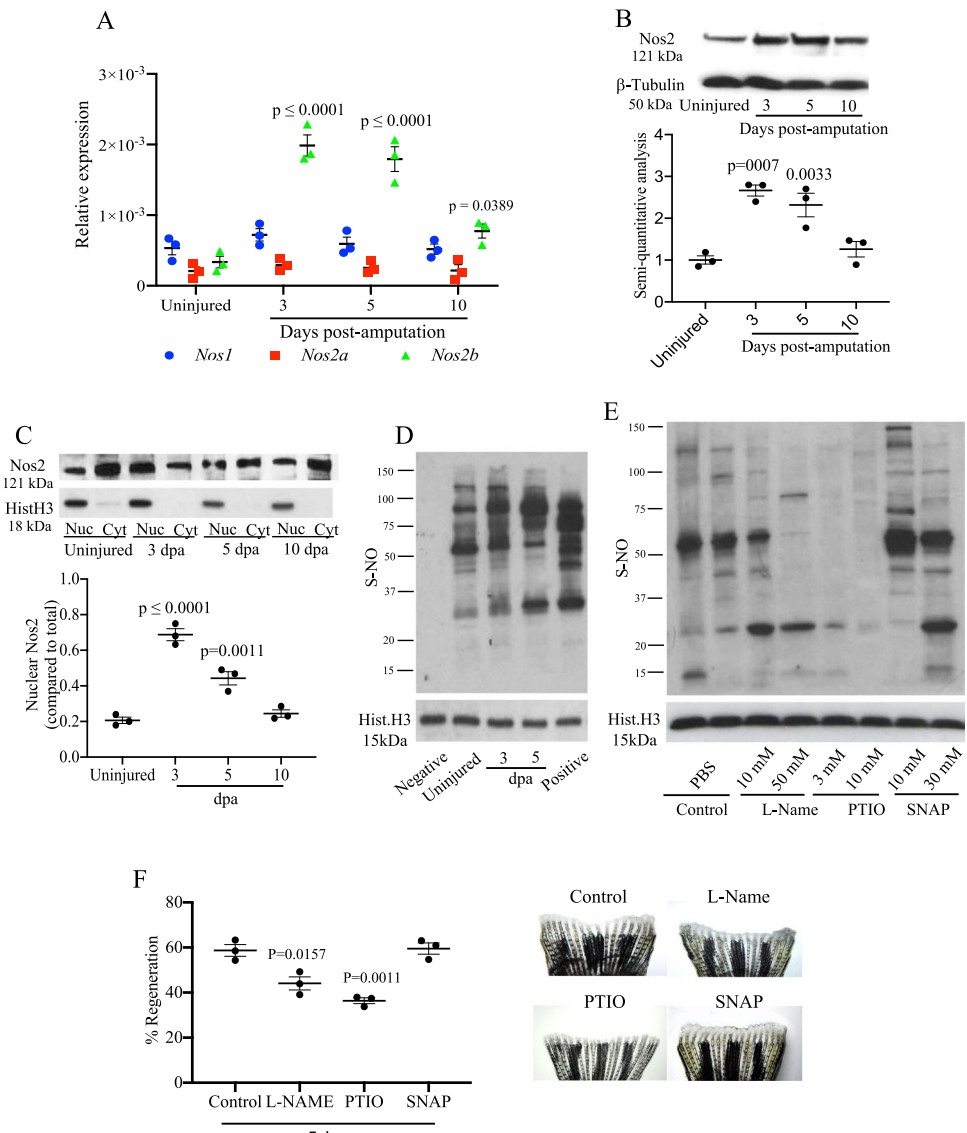

**Fig. 1 Modulation of Nos and nuclear protein S-nitrosylation during tailfin regeneration in adult zebrafish. A** Real time PCR for *nos1*, *nos2a*, and *nos2b* in tailfin at 3, 5, and 10 dpa. **B** Western blot (WB) analysis of Nos2 in tailfins at 3, 5, and 10 dpa and semi-quantitative analysis of bands. **C** WB analysis of Nos2 in nucleus and cytoplasm of tailfins at 3, 5, and 10 dpa. Semi-quantitative analysis of bands shows nuclear to cytoplasmic protein ratio. **D** WB analysis of S-nitrosylated nuclear proteins in the regenerating tailfin. S-nitrosothiols were specifically labeled with TMT. An anti-TMT antibody was used to detect S-nitrosylated proteins. Neg and Pos are respectively the negative (without ascorbic acid) and positive (with S-nitrosoglutathione) controls. **E**, **F** WB analysis of S-nitrosylated nuclear proteins in tailfin following treatment with L-NAME 10 or 50 mM, 2-Phenyl-4,4,5,5-tetramethyl imidazoline-1-oxyl 3-oxide (PTIO) 3 or 10 mM, S-Nitroso-N-acetyl-DL-penicillamine (SNAP) 10 or 30 mM, or PBS (control). The dot plot shows changes in tailfin regeneration rate, here shown at 7 dpa, following drug treatments compared to control. β-tubulin was used as loading control for total or cytoplasmic proteins. Histone H3 was used as loading control for nuclear proteins. Dpa days post-amputation. $N = 3$ biological replicates, one way ANOVA test followed by Bonferroni's multiple comparisons test was used to compare the means, *p* values shown are vs. uninjured or control, all other comparisons are not significant. Data are presented as mean values $+/-$ SEM.

healing processes, such as the epithelial–mesenchymal transition (EMT) pathway[11] and the Hedgehog pathway[12]. From the protein list (Dataset S1), we selected 31 candidates, encompassing epigenetic modifiers and transcription factors for which transient S-nitrosylation have not been previously reported and that have a human ortholog (Table S1). We proceeded to analyse these by deep mass-spectrometry, focusing initially on Kdm1A, also known as Lsd1, Kiaa061, or Aof2 (UniProtKB-F6NIA2).

**S-nitrosylation of Kdm1a is associated with a reduced binding to the CoREST complex and impaired demethylase activity on H3K4 during tailfin regeneration.** Kdm1a was the first histone lysine-demethylase to be described[13], where an amine oxidase domain mediates its FAD-dependent demethylase activity[14]. Kdm1a participates in gene repression as part of the CoREST (co-repressor for element-1-silencing transcription factor)[15] and NuRD (nucleosome remodeling and histone deacetylation)[16] co-repressor complexes mediating the demethylation of H3K4me1/me2. It also participates in gene activation in androgen receptor (Ar)-driven expression programs through demethylation of repressor marks H3K9me1/me2[17,18]. More recently, it has been demonstrated that Kdm1a is also able to demethylate lysine residues at several non-histone substrates, such as p53[19], Dnmt1 (DNA (cytosine-5)-methyltransferase 1)[20] and E2F1[21].

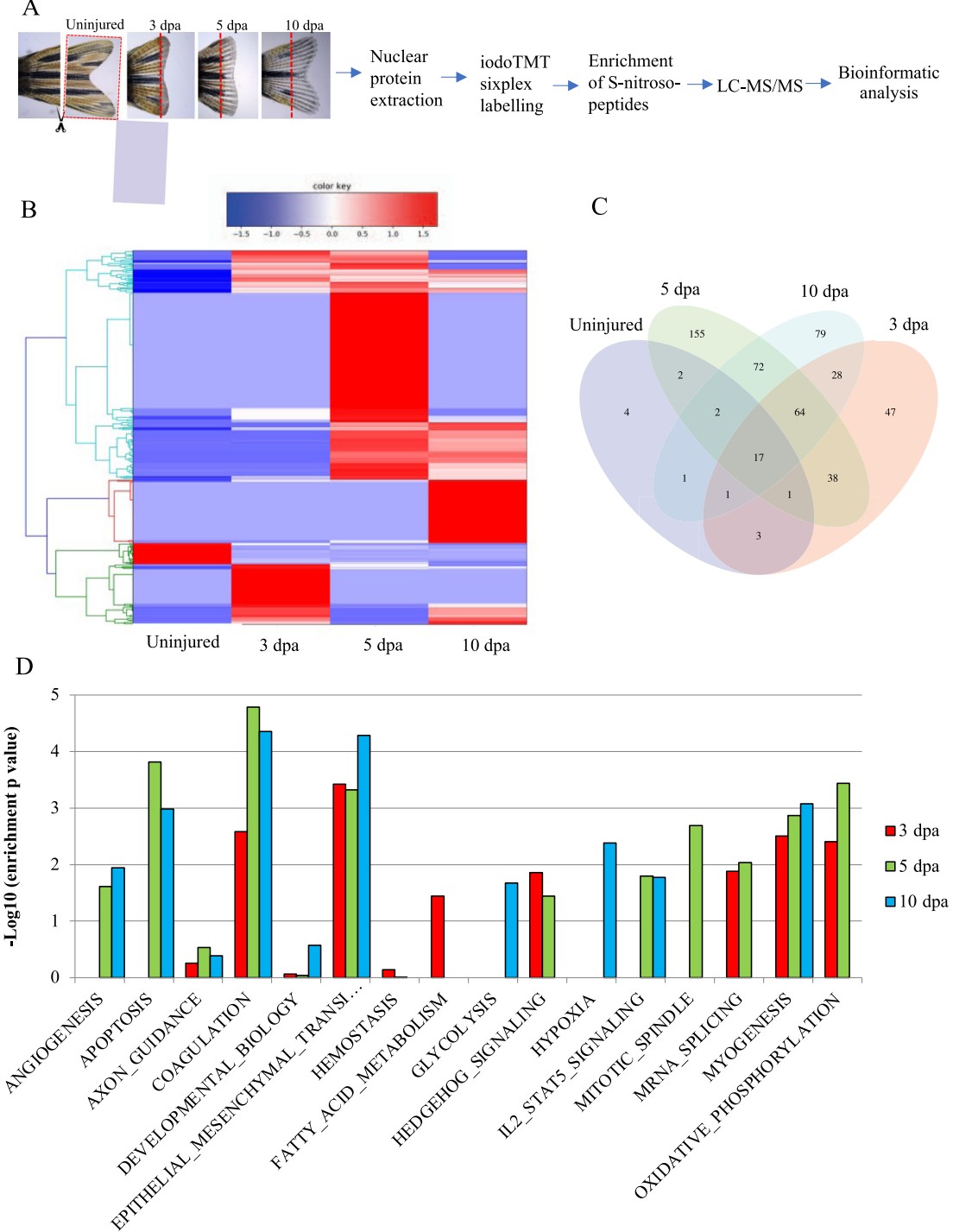

**Fig. 2 Bioinformatic analysis of the S-nitrosylome in zebrafish tailfin regeneration. A** Workflow for the analysis of the S-nitrosylome. Tailfins of zebrafish (6 months old) were amputated; regrown tissue was collected in uninjured and at 3, 5, and 10 dpa (dash red lines represent the edge of the amputation); nuclear proteins were extracted and labeled with iodoTMT, digested with trypsin, and S-nitroso-peptides enriched through anti-TMT antibody containing resin and followed by LC-MS-MS. **B, C** Hierarchical clustering heat map and Venn diagram showing the dynamic changes in the number of S-nitrosylated nuclear proteins during the regeneration compared to uninjured. **D** Hallmark pathways enrichment by the differentially expressed S-nitrosylated proteins during regeneration compared to uninjured. The significant pathways are displayed along the *x*-axis. Dpa day post-amputation. $N = 2$ biological replicates, followed by bioinformatic analysis.

Mass-spec revealed that Kdm1a, although detectable in the nucleus at all three time-points, becomes S-nitrosylated on Cysteine 334 (Cys334) at 3 and 5 dpa (Fig. 3A, B). The increase of Kdm1a S-nitrosylation (S-NO) during tailfin regeneration was confirmed by western blotting for S-NO on samples previously immunoprecipitated for Kdm1a (Fig. 3C). Zebrafish Kdm1a protein has 85% identity with the corresponding human gene with the Cys334 corresponding to Cys360 in the human ortholog (Supplementary Fig. S4A). Intriguingly, the sequence of 60 amino acids (aa) from 50 upstream to 9 downstream of the Cys334

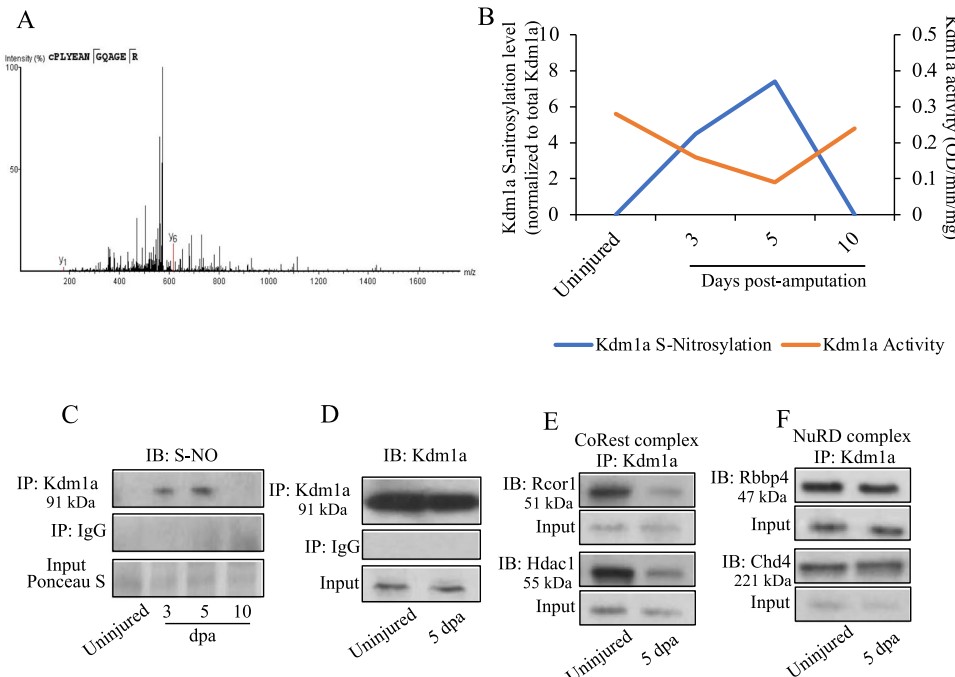

**Fig. 3 Role of S-nitrosylation of Kdm1a in tailfin regeneration in adult zebrafish. A** MS/MS fragmentation spectrum for the Cys334-containing peptide of Kdm1a. Peptide sequence is shown at the top left of the spectrum, with the annotation of the identified matched amino terminus-containing ions (b ions) in black and the carboxyl terminus-containing ions (y ions) in red. The spectrum confirms the identity of the peptides CPLYEAN and the labeled C as S-nitrosylated cysteine. **B** Line graph reporting the quantification of Kdm1a S-nitrosylation (normalized by total Kdm1a) and Kdm1a activity during tailfin regeneration. **C** Western blotting (WB) for S-nitrosylated Kdm1a in uninjured and at 3, 5, and 10 dpa. Samples were previously immunoprecipitated (IP) for Kdm1a. IP with IgG and input were used as controls. **D-F** WB for Kdm1a, CoRest and NuRD complexes components following IP with Kdm1a antibody in tailfin uninjured or injured at 5 dpa. Dpa days post-amputation. $N = 3$ biological replicates.

retains 100% identity with the human ortholog. Human KDM1a crystal structure, obtained from Protein Data Bank deposited by Tan et al.[22], is shown in Supplementary Fig. S4B.

The selectivity of Kdm1a for its main targets, H3K4 and H3K9, depends on its interaction with specific protein co-factors, including CoREST, NuRD, and Ar. PTM can modulate these interactions. Therefore, first we analysed these co-factors and their complexes in injured and uninjured tailfins. Expression of CoREST and NuRD were unchanged in injured and uninjured tailfin (Supplementary Fig. S5). The expression of *ar* was extremely low and was not different in injured versus uninjured tailfin (Supplementary Fig. S5). Having established the patterns of expression of these co-factors, and since *ar* is associated with the modulation of H3K9, we decided to focus on H3K4 which is known to be linked to CoREST and NuRD.

Hence, focusing on NuRD and CoREST, we tested the hypothesis that S-nitrosylation of Cys334 affects Kdm1a binding affinity to these complexes. To do this, we performed co-immunoprecipitation with Kdm1a antibody (Fig. 3D), followed by WB for the members of the CoREST (Rcor1 and Hdac1) (Fig. 3E) and NuRD (Rbbp4 and Chd4) (Fig. 3F) complexes. While the binding of NuRD complex was similar during regeneration compared to control, the binding to the CoREST complex was reduced.

To further assess the role of Kdm1a in the adult zebrafish, knockdown (KD) was achieved by injection of morpholino (vivo-Mo) in the retro-orbital vein. The KD efficiency as well as delivery to the tailfin was assessed by western blotting for Kdm1a (Fig. 4A). We saw clear evidence that *kdm1a* KD by vivo-Mo injection impaired tailfin regeneration compared to controls (Fig. 4B, C). A dynamic and choreographed regulation of *kdm1a* expression and activity therefore appears to be essential for tissue regeneration. These data support the role of Kdm1a as a key regulator of regeneration, likely through modulation of chromatin modifications.

During tailfin regeneration we observed a significantly reduced demethylase activity of Kdm1a in the regenerating tailfin associated with increased levels of Kdm1a S-nitrosylation (Fig. 3B). This reduced Kdm1a activity was similar to that observed in zebrafish *kdm1a* KD (Supplementary Fig. S6).

Indeed, at the same time that Kdm1a activity was reduced, there was a corresponding reduction in unmethylated H3K4 (Fig. 4D, E). By contrast, there was an increase in H3K4me1/me2, normally associated with active gene transcription (Fig. 4D, E). H3K4me3 showed a pattern similar to H3K4me2. However, it is known that Kdm1a is unable to demethylate H3K4me3, according to the chemical nature of the amine oxidation reaction catalyzed by flavin-containing amine oxidases, thus precluding H3K4me3 as a substrate[13].

In regenerating tailfins, the blastema, a pool of proliferative progenitor cells, is located at the distal region of the tailfin while cells in the proximal region undergo differentiation. To localize the region in which Kdm1a S-nitrosylation is most important, we analysed the expression of *kdm1a* gene in the distal (blastema) and proximal regions (differentiating zone) of regenerating tailfins at 3 and 5 dpa. Compared to uninjured tailfins, *kdm1a* expression was significantly lower in the blastema and only slightly reduced in the proximal differentiating zone (Supplementary Fig. S7A). We also took advantage of a publicly available single cell(sc) RNA-seq dataset obtained in the regenerating tailfin[23] in which *kdm1a* expression was detected, at low levels, with a distinct expression pattern observed in specific cell types (Supplementary Fig. S7B). For example, compared to preinjury, the *kdm1a* expression was reduced in mucosal-like cells at 1 and 2 dpa followed by an increase at 4 dpa, it was increased in mesenchymal cells and it was not expressed in hematopoietic cells during the regeneration. Overall, this data provides further evidence that modulation of S-nitrosylation, as well as modulation of gene expression, is a key

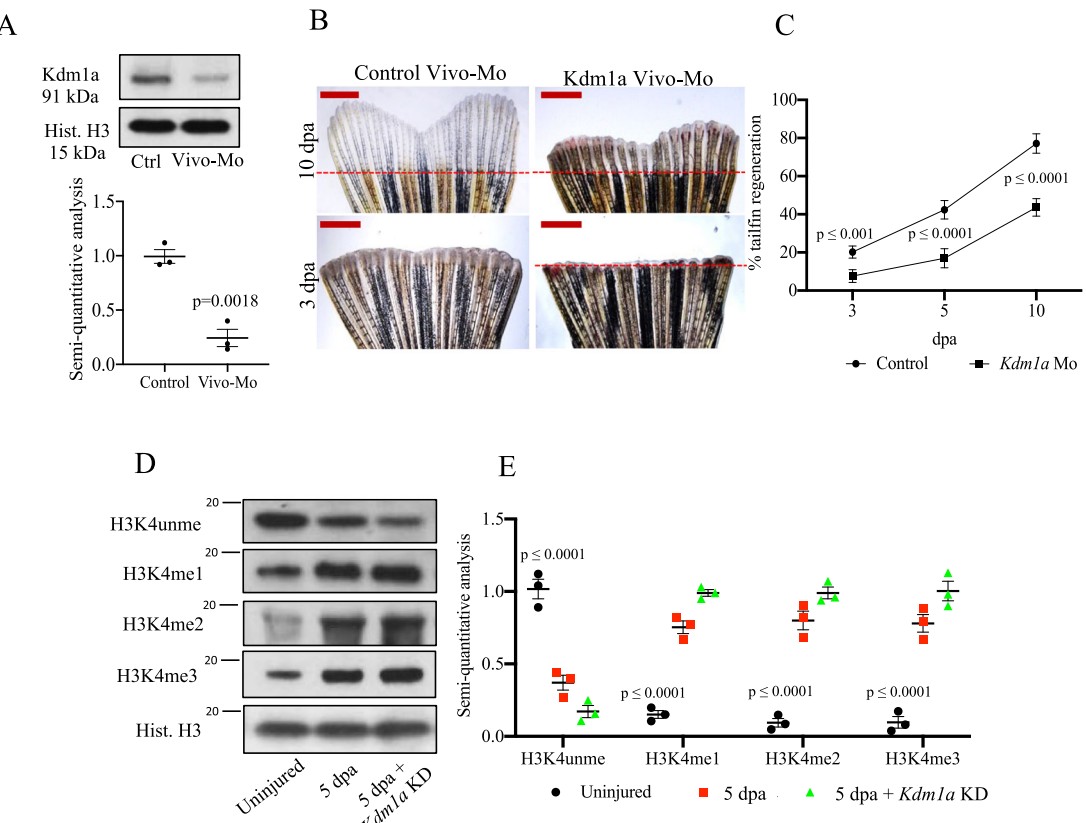

**Fig. 4 Effects of *kdm1a* knockdown in adult zebrafish. A** Western blotting (WB) analysis of Kdm1a control and morpholino KD. Dot plot shows semiquantitative analysis of bands. Two-tailed *t*-test. **B** Effects of *Kdm1a* KD on tailfin regeneration. Dashed red line represents the edge of the resection. Scale bar indicates 2 mm. **C** Line graph showing changes in tailfin regeneration rate following *kdm1a* KD. Two-way ANOVA followed by Bonferroni's multiple comparisons test. **D**, **E** WB for H3K4unme (unmethylated), H3K4me1, H3K4me2, and H3K4me3 in control uninjured, injured and injured + *kdm1a* KD at 5 dpa. Dot plot shows semi-quantitative analysis of bands. Histone H3 was used as loading control. Dpa days post-amputation. Two-way ANOVA followed by Bonferroni's multiple comparisons test, *p* values indicate comparisons of uninjured vs. other groups. N = 3 biological replicates. Data are presented as mean values +/− SEM.

mechanism regulating the functionality of Kdm1a during tailfin regeneration.

***Kdm1a* is S-nitrosylated preferentially in the endothelial cells of the regenerating tailfin.** Next, we identified S-nitrosylation associated with specific cell type. The formation of new blood vessels is a crucial process during wound healing and tissue repair, similar to embryogenesis and early growth. During tailfin regeneration, endothelial cells within the growing blood vessels sprout and invade adjacent avascular areas. By using the *Tg(fli1:EGFP)[y1]* zebrafish line, where endothelial cells (EC) fluoresce green (GFP+ cells), we detected the formation of new blood vessels in the regenerating tailfin at 2 dpa that become a dense vascular plexus by 3 dpa (Fig. 5A) consistent with previous observations[24]. We isolated EC from these tailfins and performed FACS analysis. We found an increase in the percentage of EC (GFP+) from 4.3 ± 0.3% to 6.1 ± 0.4%, in the uninjured versus injured tailfin, respectively (Fig. 5B, C and Supplementary Fig. S7C). We FACS purified GFP+ (EC) and GFP− (non-EC) from injured and uninjured tailfins, extracted RNA and performed real time PCR. For *kdm1a*, we found no difference in gene expression between groups (Supplementary Fig. S8A). We also analysed the expression of CoREST (*rcor1* and *hdac1*) and NuRD (*rbbp4* and *chd4*) factors and did not detect significant differences between groups (Supplementary Fig. S8B–E). In response to stress or injury, EC upregulate *inos*, with consequent NO production[25]. We found an upregulation of *nos2b* compared to *nos2A* and *nos1*

genes in the EC of injured tailfins compared to uninjured (Supplementary Fig. S8F–H). Therefore, we investigated if S-nitrosylation, more specifically of Kdm1a, occurs in EC during tailfin regeneration. We observed an overall increase in S-nitrosylated proteins in the EC of regenerating tissue compared to EC from uninjured tissue (Fig. 5D), with no change in S-nitrosylated proteins in GFP− cells from injured versus uninjured tailfins. Then, we specifically analysed S-nitrosylated Kdm1a in EC cells from injured and uninjured tailfins at 5 dpa, including an injured group treated with the nitric oxide (NO) scavenger PTIO. While the expression of total *kdm1a* did not change in the three groups, S-nitrosylated Kdm1a was detectable in EC from regenerating tailfin only (Fig. 5E), while it was abolished in the group treated with PTIO. The increased S-nitrosylation of Kdm1a in EC of the regenerating tailfin was associated with a significant increase in vessel density compared to the other two groups (Fig. 5F).

**S-nitrosylation of Kdm1a is associated with increased H3K4me2 marks for endothelial genes during tailfin regeneration.** As shown in Fig. 4D, E, H3K4me2, a target of Kdm1a, is increased during regeneration associated with the reduced demethylase activity of the S-nitrosylated form of Kdm1a. Methylated H3K4 is associated with active gene transcription. Accordingly, we reasoned that an increased occupancy of H3K4me2 would result in increased levels of proangiogenic factors. To address this hypothesis, and investigate whether the increased vessel density observed during regeneration was correlated to S-nitrosylation of Kdm1a, we

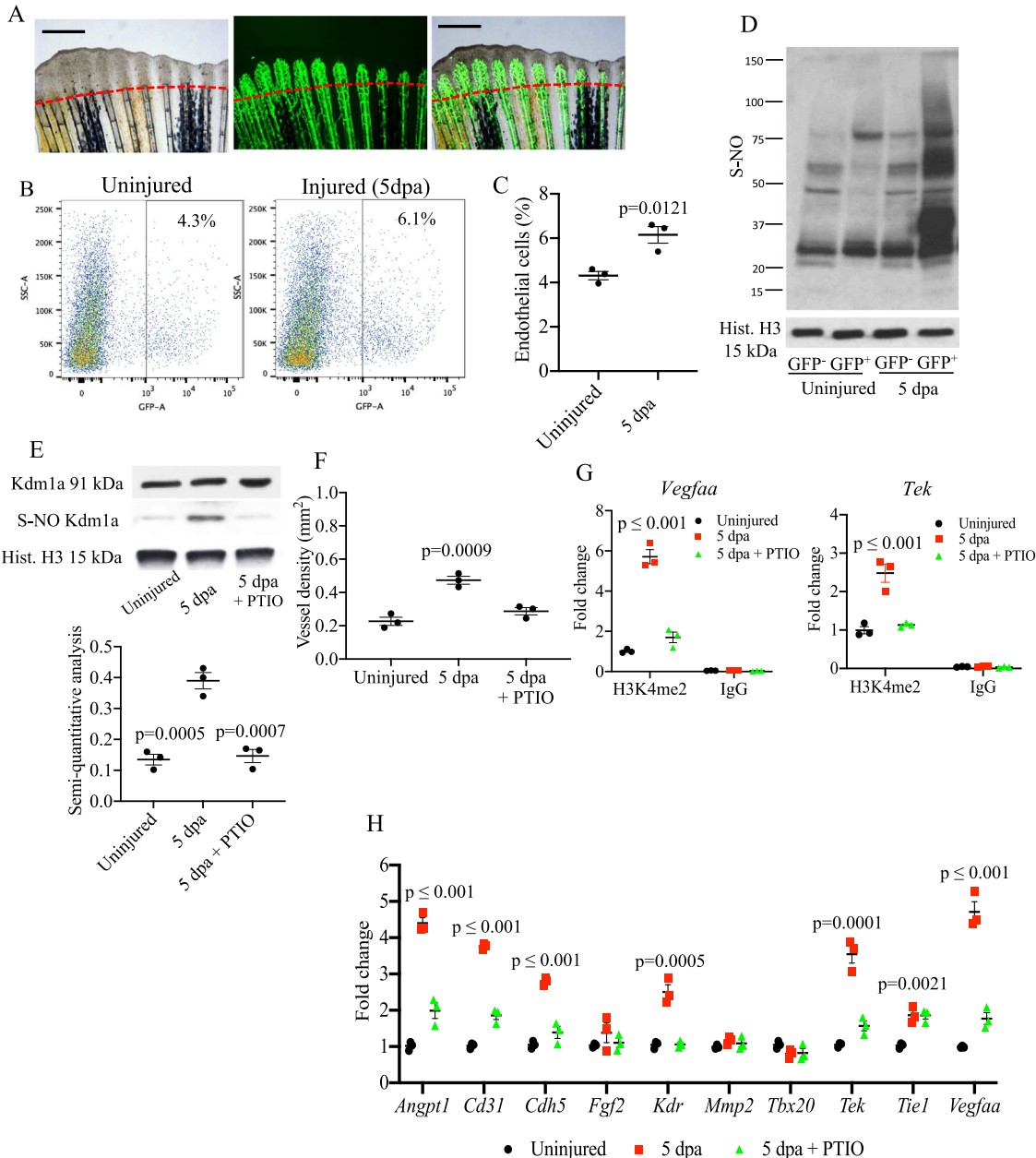

**Fig. 5 Analysis of S-nitrosylation in endothelial cells during tailfin regeneration. A** Brightfield and fluorescence images of *Tg(fli1:EGFP)$^{y1}$* zebrafish tailfin at 3 days post-amputation (dpa) showing formation of new vessel branches (GFP signal). Scale bar measures 500 μm. **B** FACS plot of GFP$^+$ and GFP$^-$ cells in the tailfin in control and during regeneration were separated by FACS. **C** Quantification of GFP+ cells as shown in FACS plots. Two-tailed *t*-test. **D** Western blotting (WB) of total S-nitrosylated proteins in zebrafish tailfin endothelial (GFP+) cells. **E** WB of Kdm1a and S-nitrosylated Kdm1a in endothelial (GFP+) cells control, injury and injury + PTIO (NO scavenger) 10 mM. Dot plot shows semi-quantitative analysis. *p* values vs. 5 dpa group. **F** Vessel density analysis in *Tg(fli1:EGFP)$^{y1}$* zebrafish tailfin uninjured, injured and injured + PTIO 10 mM, measured as total length of vessels. **G** ChIP-PCR analysis in GFP$^+$ cells isolated from the regenerating tailfin showing H3K4me2-binding complex with *vegfaa* and *tek* promoters. Rabbit IgG were used as a negative control. **H** Real time PCR analysis for endothelial genes in GFP+ cells from zebrafish control, injury and injury + PTIO 10 mM. Histone H3 was used as loading control. One-way ANOVA followed by Bonferroni's multiple comparisons test, *p* values indicate comparisons vs. uninjured. $N = 3$ biological replicates. Data are presented as mean values +/− SEM.

performed Chromatin Immunoprecipitation using a ChIP-grade antibody specific for H3K4me2, followed by PCR for 10 well-known proangiogenic factors, including *kdr, vegfaa, fgf2, angpt2, tek, tie1, cdh5, cd31, mmp2,* and *tbx20*. ChIP-PCR showed that the occupancy of H3K4me2 on *vegfaa* and *tek* in endothelial cells from injured tailfins was significantly higher compared to injured tailfins treated with PTIO and higher than uninjured tailfins (Fig. 5G). Furthermore, real time PCR showed that expression of 7 out of 10 proangiogenic factors, including *vegfaa* and *tek*, were significantly

increased during regeneration (Fig. 5H), whereas they were unchanged following treatment with PTIO, again suggesting that nitric oxide was driving or facilitating angiogenesis via a mechanism involving S-nitrosylation.

Overall, these findings show that endothelial cells of the regenerating tissue are a key cell target for protein S-nitrosylation after injury and that Kdm1a is specifically S-nitrosylated in these cells. S-nitrosylation of Kdm1a reduces Kdm1a demethylase activity and, as a consequence, more H3K4me2 will accumulate in

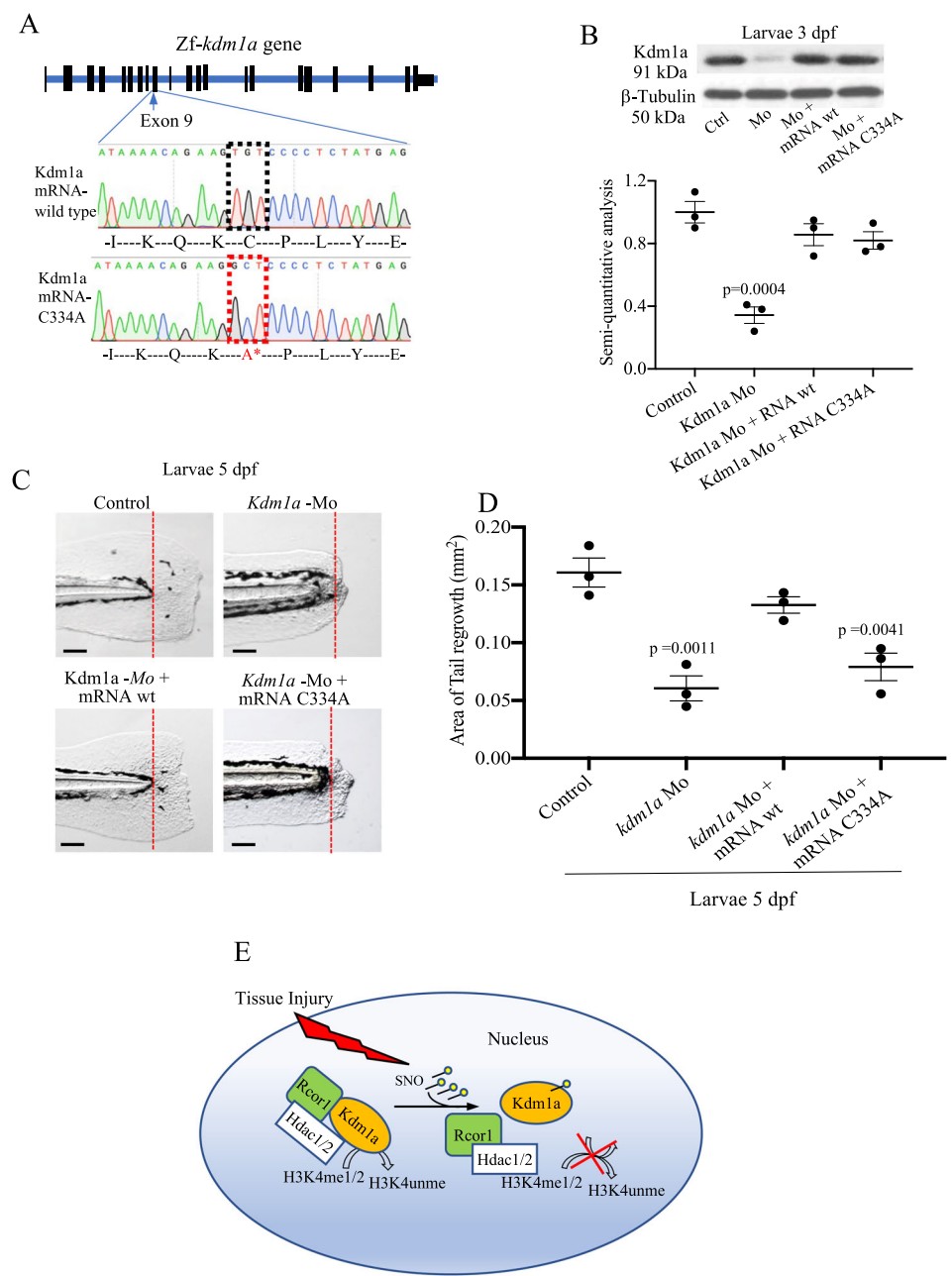

**Fig. 6 Modulation of Kdm1a S-nitrosylation during tailfin regeneration. A** *Kdm1a* mRNA C334A was generated by site-directed mutagenesis, replacing the aa Cys334 with Ala. **B–D** Zebrafish embryos injected with *kdm1a* morpholino (Mo), or co-injected with *kdm1a* Mo with *kdm1a* mRNA C334A or wild type. **B** Western blotting and semi-quantitative analysis showed the effective knockdown and rescue of *kdm1a* following the different treatments. β-tubulin was used as loading control. **C, D** Images and dot plot of tailfin regeneration following *kdm1a* modulation (*p* values vs. control). Scale bar measures 100 μm. **E** Working model. Tissue injury promotes the S-nitrosylation of the Cys334 of Kdm1a. S-nitrosylated Kdm1a detaches from the CoRest complex and loses its demethylase activity on H3K4. One-way ANOVA followed by Bonferroni's multiple comparisons test, *p* values indicate comparisons vs. control. *N* = 3 biological replicates. Data are presented as mean values +/− SEM.

the genome. In particular, we found increased occupancy of H3K4me2 on *vegfaa* and *tek*.

This does not exclude, however, that other S-nitrosylated proteins among those arising from the mass-spec dataset could be implicated in endothelial cells and vascular regrowth as well as in other cell types.

Next, to investigate the effects of the absence of the Kdm1a C334 S-nitrosylation site on tailfin regeneration we synthesized, by site-directed mutagenesis, a mutant variant of Kdm1a mRNA where the cysteine 334 was substituted with alanine (C334A), that cannot be S-nitrosylated (Fig. 6A). These experiments assessed

the ability of this mutated Kdm1a mRNA to rescue the morpholino (Mo) phenotype compared to wild-type mRNA. However, contrary to the MO oligo that is a stable molecule, not degraded by the nucleases, the mRNA is prone to degradation by nucleases, well before reaching the tailfin, i.e., the tissue under study. Therefore, we decided to perform these studies in the zebrafish embryos where the cytoplasmic bridges connecting the early embryonic cells allow rapid diffusion of mRNA into the cells, resulting in fast and ubiquitous delivery[26].

Before the morpholino (Mo) studies, real time PCR analysis confirmed *kdm1a* expression during development, with a slight

reduction from 24 to 120 hpf (Supplementary Fig. S9A). This finding was also useful to optimize the dose of Mo to inject. Mo experiments were conducted according to the guidelines[26,27] using several controls to assess morpholino specificity for *kdm1a*. First, we injected *kdm1a*-targeted Mo or the mismatch (control), using an optimized dose of 0.8 ng per egg. The effective *kdm1a* KD was confirmed by western blotting (Fig. 6B). The survival rate of *kdm1a* KD embryos at 120 hpf was approximately 80% compared to 90% in controls (Supplementary Fig. S9C). *Kdm1a* KD embryos did not show gross abnormalities compared to control (Supplementary Fig. S9D), however we found reduced blood flow velocity (Supplementary Fig. S9E) and reduced expression of *gata1* (Supplementary Fig. S9F), a red blood cell marker, consistent with a previous report[28]. To confirm these effects and to exclude sequence-specific off-target effects we used a second Mo with non-overlapping sequence to compare phenotypes with the first Mo. Indeed, both Mo injections induced *kdm1a* knockdown and produced a comparable phenotype. To further confirm the specificity, we co-injected half-doses of each Mo, such that the phenotype was only just apparent with each Mo alone, but with clear additive effects on phenotype when co-injected. Specifically, the two *kdm1a*-targeted antisense Mo were co-injected each at 1/2 dose (0.4 ng per egg). While the phenotype was apparently unaffected with 1/2 dose of each oligo injected alone, the co-injection of half doses together produced phenotypic effects similar to those produced by a single oligo at full dose (0.8 ng per egg) (Fig. 6B and Supplementary Fig. S9). These additive effects of low doses of two antisense oligos strongly supports a *kdm1a*—specific effect. We did not find an increase in expression of *tp53*, well-known off-target effect, at the dose of Mo used in this study, therefore we did not co-inject *tp53*-Mo for *tp53* gene silencing (Supplementary Fig. S9B). We observed that the regeneration of the tailfin was significantly reduced in *kdm1a* KD embryos compared to control (Fig. 6C, D), with a phenotype penetrance of >70%. The phenotype of *kdm1a* morphants was rescued by co-injecting *kdm1a* mRNA indicating that the observed effects were specific for *kdm1a*. This also clearly shows a key role of *kdm1a* in tailfin regeneration. However, *kdm1a* mRNA C334A did not rescue tailfin regeneration (Fig. 6C, D and Supplementary Fig. S9). This further confirmed the crucial role of S-nitrosylation of this specific cysteine residue on Kdm1a in modulating tailfin regeneration.

## Discussion

Following stress or injury, *inos* is activated within local somatic cells[29], leading to protein S-nitrosylation, the covalent attachment of a NO group to the thiol side chain of the cysteine. This mechanism has emerged as a dynamic, post-translational regulatory mechanism for many classes of proteins[30]. We found increased expression of *inos* in the nuclei of the regenerating tailfin of the zebrafish, and over 500 nuclear proteins that became S-nitrosylated during regeneration. Of these, we identified a key role for Kdm1a. We demonstrated a strong link between Kdm1a S-nitrosylation and its role in histone demethylation and ultimately in regeneration of the zebrafish tailfin. Endothelial cells, where *inos* expression increases after stress or injury, seems to be the main cell target where S-nitrosylation, including of Kdm1a, mostly occurs. The essential role of Kdm1a in hematopoiesis has been shown in vitro[31,32] and in vivo[33,34]. As such, understanding the molecular mechanisms underpinning the action of Kdm1a are currently being investigated to find specific inhibitors that could be harnessed as a therapeutic strategy in cancer[35]. According to our bioinformatic analysis of single cell-RNAseq, *kdm1a* was not detected in hematopoietic cells of the zebrafish regenerating tailfin. Nonetheless, given the role of the Kdm1a in hematopoiesis we cannot exclude the possibility that

impaired fin regeneration in *kdm1a* knockdown zebrafish could, at least partially, derive from systemic effects associated with altered immune cell function.

In our study, the S-nitrosylated form of Kdm1a has a reduced demethylase activity on H3K4 that results in a corresponding increase in H3K4me2 that, by turn, promotes the expression of proangiogenic genes. This is the first study to show the importance of the S-nitrosylation of nuclear proteins in tissue regeneration and potentially opens up new therapeutic avenues (Fig. 6E).

While many interesting candidate S-nitrosylated nuclear proteins were identified (Dataset S1 and Table S1), we focused on Kdm1a. However, we predict that it is likely that Kdm1a is not the only rate limiting factor. It is likely that modulating the expression and S-nitrosylation of other candidates could result in similar effects on regeneration, possibly by acting via the same cells and pathways identified here but possibly on other cell types and pathways. This would depend on the effects that S-nitrosylation has on molecular networking of each protein candidate. For example, S-nitrosylation of Hexim1 (hexamethylene bisacetamide inducible protein 1) could affect its binding in the P-TEFB complex[36] and potentially inhibits the Cdk9 kinase activity and the transcription of genes specific to regeneration.

Aberrant levels of protein S-nitrosylation have been implicated in a number of diseases, including heart disease, diabetes, cancer, neurological disorders, chronic degenerative diseases, and inflammatory disorders (reviewed in ref. [37]), regeneration. Furthermore, very little is known about two important aspects of this process, firstly the extent and the dynamics of S-nitrosylation of *nuclear* proteins in human disease and secondly the role of S-nitrosylation of nuclear proteins during tissue repair and regeneration[38]. Our paper is the first to examine the mechanisms and potential role of S-nitrosylation of nuclear proteins during tissue regeneration in vivo and provides several important observations that implicate S-nitrosylation of nuclear proteins in regeneration.

## Methods

**Ethical approval.** This work complied with all relevant ethical regulations for animal testing and research. Animals were housed and all experiments were carried out in accordance with the recommendations of the Institutional Animal Care and Use Committee at the Houston Methodist Research Institute, and with the United Kingdom Animals (Scientific Procedures).
Act 1986 at the Queens Medical Research Institute research facilities.

**Zebrafish aquaculture and husbandry.** Adult zebrafish—wild-type *Wik* and *Tg(nfkb:EGFP)^{nc1}* strains—were maintained according to standard procedures. Fish were kept at 28 °C under a 14/10 h light/dark cycle and fed with dry meal (Gemma Micro, Westbrook, ME) twice per day. Embryos were obtained by natural mating and kept in E3 embryo medium at 28.5 °C. Surgical procedures were performed under anesthesia with Tricaine (also named MS-222, Sigma-Aldrich, St Louis, MO, cat. E10521) 0.02 mg/mL on embryos and 0.05 mg/mL in adult zebrafish.

**Zebrafish tailfin amputation and regeneration.** Caudal tailfin amputation surgeries were performed as previously described[39]. Briefly, fish were anaesthetized and amputations were made by using a sterile razor blade, removing half of the tailfin. At day 3, 5, and 10 post-amputation (dpa) (uninjured tailfin was used as control) the regrown tissue was carefully resected and immediately processed for nuclear protein extraction. Total regeneration was measured as previously described[40]. Briefly, fin images were collected before amputation and time points after amputation. The new tissue area (in pixels) of the caudal fin from the new distal fin edge to the amputation plane was quantified in each fish using Image J software. The percentage of regeneration for each fin at each time point was defined as percentage of regeneration = $100 \times$ (regenerated tissue area/original fin area amputated). The collection of the distal (blastema) and proximal (regenerating) regions of the tailfin tissue for PCR analysis was performed under a fluorescence stereomicroscope Leica M205. The *Tg(fli1:EGFP)* zebrafish was used and the distal ends of the newly forming (GFP+) vessel branches was used as boundary to separate by dissection the two regions. Then, we placed the tissues in an Eppendorf tube and immediately extracted RNA that was used for PCR analysis.

**Blood vessel density.** At 5 dpa, the adult *Tg(fli1:EGFP)* zebrafish were anesthetized. Then, fish were transferred on a wet sponge, previously soaked in tank water, to keep

the zebrafish skin moist during imaging. Images of the caudal fin including the regenerated tissue were captured under a fluorescence stereomicroscope Leica M205 equipped with a camera. The images were collected with a Leica LAS X software and analysed by Image J software. The second and third rays from the dorsal edge of the fin were used for measurement of vessel area and reported as mm[2].

### Protein extraction

*Extraction of nuclear proteins.* Nuclear proteins were extracted from the caudal fin tissue using the NE-PER Extraction Reagents kit (Thermo Fisher Scientific, San Jose, CA, cat. 78833) according to the manufacturer's instructions, supplemented with protease inhibitor cocktail. In brief, regenerating tailfin tissue was resected, cut into small pieces and placed in a microcentrifuge tube. Then, tissue was washed three times with chilled PBS, centrifuged at $500 \times g$ for 1 min and supernatant discarded. Using a motor-driven pestle (Sigma-Aldrich, St Louis, MO, cat. Z359971), tissue was homogenized in solution CER I, that breaks plasma membrane but not nuclei, added with protease/phosphatase inhibitors cocktail (1:100, Thermo Fisher Scientific, San Jose, CA, cat. 78442). The tube was vigorously vortexed for 30 s and put on ice for 10 min. Then, chilled CER II was added to the tube, vortexed for 10 s and incubated on ice for 1 min. The tube was centrifuged for 5 min at $16,000 \times g$ and the supernatant, containing cytoplasmic proteins, was transferred to a clean pre-chilled tube and stored at $-80\,°C$. The pellet, containing nuclei, was resuspended in chilled NER solution, vortexed on the highest setting for 15 s. The sample was placed on ice for 45 min, vortexed for 20 s every 10 min. Then, the tube was centrifuged at $16,000 \times g$ for 10 min and the supernatant, containing nuclear extract, immediately transferred to a clean pre-chilled tube.

*Extraction of total proteins.* Zebrafish embryos were euthanised with an overdose of tricaine, washed three times in PBS and homogenized with a motor-driven pestle (Sigma-Aldrich, St Louis, MO, cat. Z359971) in 100 mL RIPA buffer (25 mmol/L Tris-HCl pH 7.6, 150 mmol/L NaCl, 1% NP-40, 1% sodium deoxycholate, 0.1% SDS), supplemented with protease/phosphatase inhibitors. The lysate was kept on ice for 40 min. Then, the tube was centrifuged at $3000 \times g$ for 5 min and the supernatant transferred to a clean pre-chilled tube.

In both nuclear and total proteins extraction, bicinchoninic acid (BCA) protein assay (Thermo Fisher Scientific, San Jose, CA, cat. 23225) was used to measure protein concentration.

### Labeling of protein S-nitrosothiols

Labeling of S-nitrosothiols in nuclear proteins was achieved using Iodoacetyl Tandem Mass Tag (iodoTMT) kit (Thermo Fisher Scientific, San Jose, CA, cat. 90103). First, nuclear protein extracts were acetone-precipitated at $-20\,°C$ for 2 h, then centrifuged at $15,000 \times g$ for 10 min and the pellet solubilized in 500 mL HENS buffer (Thermo Fisher Scientific, San Jose, CA, cat. 90106) at a protein concentration of 1 mg/mL. Equal amounts of nuclear protein from each sample were iodoTMT-labeled. To generate a positive control sample, an aliquot of protein from control was added with 200 μM S-nitrosoglutathione (GSNO, Sigma-Aldrich, St Louis, MO, cat. N4148) for 30 min at room temperature (RT). Experimental samples were incubated for 30 min at RT after adding MMTS (10 μL of 1 M) to block-free cysteine thiols. Then, proteins were precipitated with pre-chilled acetone (1 mL per sample) at $-20\,°C$ for 2 h to remove MMTS. Samples were centrifuged at $10,000 \times g$ for 10 min at 4 °C, the pellet resuspended in 500 mL of HENS buffer and to each was added 5 mL of iodoTMT reagent, previously dissolved in liquid chromatography/mass spectrometry (LC/MS)-grade methanol, and 10 μL of 1M sodium ascorbate (Sigma-Aldrich, St Louis, MO, cat. A4034). As a negative control, 10 μL of ultrapure water instead of sodium ascorbate was added in a protein sample. All samples were incubated for 1 h at 37 °C, protected from light. The reaction was quenched by adding 20 μL of 0.5M DTT and incubated for 15 min at 37 °C, protected from light. All combined samples labeled with iodoTMT sixplex were combined, added with six volumes of pre-chilled acetone and incubated at $-20\,°C$ overnight. The sample was centrifuged at $10,000 \times g$ for 10 min at 4 °C and the pellet dissolved in 3 mL HENS buffer. Then, 100 μL of 0.5 M iodoacetamide were added and the sample incubated at 37 °C for 1 h protected from light. Sample was precipitated with pre-chilled acetone and the pellet allowed to dry for 10 min.

### Protein digestion for mass-spec analysis

The pellet was dissolved in 50 mM ammonium bicarbonate (Sigma-Aldrich, St Louis, MO, cat. 09830) and digested using trypsin enzyme (GenDepot, cat. T9600) at 37 °C overnight. The peptide mixture was acidified using 10% formic acid and dried using a vacuum concentrator (Thermo Fisher Scientific, San Jose, CA, cat. SPD120).

### Enrichment of iodoTMT-labeled S-nitrosopeptides

The anti-TMT Antibody Resin (Thermo Fisher Scientific, San Jose, CA, cat. 90076) was washed three times with Tris Buffered Saline (TBS) (Thermo Fisher Scientific, San Jose, CA, cat. 28358). Previous labeled and lyophilized peptides were resuspended in TBS (a small portion of unfractionated sample was stored for direct analysis of the non-enriched samples). Then, peptides were added to the anti-TMT resin (100 μL of settled resin for every 1 mg of iodoTMT Reagent-labeled peptides) and incubated at RT for 4 h. Finally, the resin was washed three times (5 min per wash) with TBS and then three times with water. The sample was eluted with TMT Elution Buffer

(Thermo Fisher Scientific, San Jose, CA, cat. 90104). The eluate was frozen and lyophilized, using a vacuum concentrator and then the sample resuspended in a solution of 5% methanol/0.1% formic acid. Then, 1–5 μL of sample were injected directly onto an LC-MS/MS system.

### LC/MS-MS

The mass spectrometry analysis of S-nitrosopeptides was carried out on a nano-LC 1200 system (Thermo Fisher Scientific, San Jose, CA) coupled to Orbitrap Fusion™ Lumos ETD (Thermo Fisher Scientific, San Jose, CA) mass spectrometer. The peptides were loaded onto a Reprosil-Pur Basic C18 (1.9 μm, Dr. Maisch GmbH, Germany) pre-column of 2 cm × 100 μm and in-lined an in-housed 5 cm x 150 μm analytical column packed with Reprosil-Pur Basic C18 beads. The peptides were separated using a 75 min discontinuous gradient of 5-28% acetonitrile/0.1% formic acid at a flow rate of 750 nL/min. The eluted peptides were directly electro-sprayed into the mass spectrometer. The instrument used the multi-notch MS3-based TMT method. The full MS scan was performed in Orbitrap in the range of 375–1500 $m/z$ at 120,000 resolution followed by ion trap CID-MS2 fragmentation at precursor isolation width of 0.7 $m/z$, AGC of $1 \times 10^4$, maximum ion accumulation time of 50 ms. The top ten fragment ions from MS2 was selected for HCD-MS3 with isolation width of 2 $m/z$, AGC $5 \times 10^4$, collision energy 65%, maximum injection time of 54 ms. The RAW file from mass spectrometer was processed with Proteome Discoverer 2.1 (Thermo Fisher Scientific, San Jose, CA) using Mascot 2.4 (Matrix Science) with percolator against Zebrafish Uniprot database. The precursor ion tolerance and product ion tolerance were set to 20 ppm and 0.5 Da, respectively. Variable modifications of oxidation on Methionine (+15.995 Da) and iodoTMT tag (+329.2266 Da) on cysteine residues was used. The general quantification in consensus workflow used unique and razor peptides with top three peptides for area calculation, while reporter quantification used co-isolation threshold of 50 and average reporter S/N threshold of 10. The assigned peptides and PSMs were filtered at 1% FDR.

### Bioinformatic analysis

The computational detection strategy identifies peptides exhibiting iodoTMT tag modifications. Proteome Discoverer software (Thermo Fisher Scientific, San Jose, CA) was used to search MS/MS spectra against the Zebrafish UniProt database (Danio rerio; UP000000437) using Mascot 2.3 search engine. The iodoTMTsixplex quantification method within Proteome Discoverer software was used to calculate the reporter ratios with a mass tolerance ±10 ppm. Search algorithms, including MS-Fragger was also used in the analysis. Hierarchical clustering of S-nitrosylated protein expression heatmap was conducted using MEV based on Pearson correlation distance metric and the average linkage method. Ingenuity pathway analysis (IPA, Ingenuity systems Qiagen, Redwood City, CA) was used to assess Gene Ontology (GO) and IPA analysis to explore the function of differential S-nitrosylated proteins. Enrichment $Q$ values of S-nitrosylated protein pathways were defined based on EdgeR FDR cutoff 1e−5. The GO category was classified by Fisher's exact test, and the $p$-value was corrected by the false discovery rate (FDR) calculation.

The scRNA-seq of the regenerative caudal fin was analysed following the methods described in the manuscript[23]. The matrix count from cell ranger were obtained on GEO database accession GSE137971. Downstream analysis was performed on R using the package Seurat. Clustering analysis was performed on the integrated dataset and we found six clusters. These clusters were annotated as Superficial/intermediate/basal epithelial, mucosal-like, hematopoietic and mesenchymal based on markers described in Hou et al.[23]. Differential gene expression between the different time points compared to pre-injury was done using the function FindMarkers from Seurat (based on Wilcoxon test followed by Bonferroni correction). Violin Plot of *kdm1a* expression was obtained using the Vln Plot function of Seurat.

### Chromatin immunoprecipitation (ChIP)-PCR assay

ChIP assay was performed following the manufacturer's instructions (Cell Signaling Technology, Beverly, MA). Briefly, 15 tailfins of adult zebrafish per group were disaggregated in single cells as described above. DNA and protein were crosslinked by 1% formaldehyde. Chromatin was isolated and digested with Micrococcal Nuclease. Then, the DNA-protein complex was precipitated with control IgG or antibody against H3K4me2 (rabbit polyclonal, ChIP grade) overnight at 4 °C and protein A/G conjugated magnetic beads for 1 h. Cross-links were reversed. The extracted DNA was used as template for PCR amplification of the targeted promoter region. The extracted DNA from unprecipitated DNA-protein complex was used as input. The promoter regions of ten genes known to be involved in neoangiogenesis (*kdr, vegfaa, cdh5, tek, tie2, tbx20, fgf2, angpt2, mmp2,* and *cd31*) were identified in ENSEMBL. The gene sequence up to 600 bp upstream of the TSS was validated this sequence on genome.ucsc.edu to confirm it was upstream of our gene of interest. We also looked for the presence of CpG islands and TATA box. Hence, we designed four couples of primers using Primer Blast for each gene that matched in this region and around the TSS, and that could generate amplicons which size was no more than 120–130 bp to allow both primers to find their target on one fragment of ChIP DNA, if present. In silico PCR (UCSC) was used to confirm that primers matched our region of interest and the amplicon size, and then primers were further validated by PCR using genomic DNA.

**Quantification of Kdm1a demethylase activity.** Kdm1 Activity Colorimetric Kit (Abcam, Cambridge, UK, cat. ab113459) was used to quantify Kdm1 activity. Nuclear proteins were extracted from the regenerating caudal fin as shown above and an input of 10 μg per sample was used for the enzymatic analysis. The experiment was run in triplicate. A standard curve was prepared with Kdm1a assay buffer and assay standard solution, containing demethylated histone H3K4, diluted at concentration between 0.2 and 5 ng/μL. Sample wells were added with Kdm1a assay buffer, Kdm1a substrate (containing di-methylated histone H3K4) and 10 μg of nuclear extract. No nuclear extract was added in blank wells. The strip-well microplate was covered with adhesive film to avoid evaporation, and incubate at 37 °C for 2 h. At this stage, active Kdm1a binds to the substrate and removes methyl groups from the substrate. Then, the reaction solution was removed and each well washed three times with wash buffer. Capture antibody, that recognizes Kdm1a-demethylated products, was added to each well, the strip-well was covered with aluminum foil to protect from light and incubated at RT for 60 min. Antibody solution was removed and each well washed three times with wash buffer. Then, detection antibody was added to each well, covered again with aluminum foil and incubated at RT for 30 min. Detection antibody solution was removed and each well was washed four times with wash buffer. Developer solution was added and the microplate incubated at RT for 10 min protected from light. In presence of methylated DNA, the solution will turn blue. At this point, stop solution was added to each well to quench the enzymatic reaction. Absorbance was read on a microplate reader Infinite M1000 (Tecan, Männedorf, Switzerland) at a wavelength at 450 nm with an optional reference wavelength of 655 nm. The activity of Kdm1a enzyme is proportional to the optical density (OD) intensity measured. Accordingly, Kdm1a activity was calculated using the following formula:

$$\text{Kdm 1a activity}(\text{OD}/\min/\text{mg}) = \text{Sample OD-Blank OD}/(\text{Protein amount}(\mu g) \times \min)$$

**Kdm1a suppression.** The knockdown (KD) of kdm1a gene (NM_001242995.1) in zebrafish was achieved by injection of antisense morpholino (Mo) (Gene Tools, Philomath, Oregon) oligo targeting the mRNA AUG translational start site (sequence 5′-TTGGACAACATCACAGATGACAGAG-3′). A 5-base pair mismatch Mo (sequence 5′-TTGCAgAACATgACAcATcACAGAG-3′) was used as control to detect possible off-target effects. A second antisense oligo targeting i3e4 splice junction of kdm1a, sequence 5′- CTACACCTGAGAAACCCAAC ATTTC-3′ was used to corroborate data obtained with MOs that block translation.

Using a standard microinjector (IM300 Microinjector; Narishige, Tokyo, Japan), an optimized dose of 0.4 ng (0.5 nL bolus) of morpholino placed in a pulled glass capillary was injected in each embryo at 1–2 cell stage, just beneath the blastoderm.

For KD of kdm1a in adult zebrafish, a vivo-Mo version was used, where the standard Mo is bound to a synthetic scaffold containing guanidinium groups as a delivery moiety in adult tissues. An antisense vivo-Mo that targets human b-globin intron mutation 5′-CCTCTTACCTCATTACAATTTATA-3′ was used as negative control (Gene Tools LLC, Philomath, Oregon). Adult zebrafish were anesthetized in tricaine 0.05 mg/mL, and 2 μL of 0.1 mmol/L vivo-Mo solution, previously loaded in a glass capillary, was injected into the retro-orbital vein, as previously described[41], on days 10, 8, 6, 4, 2 and 0 before tailfin amputation.

**kdm1a construct.** A construct with kdm1a gene of Danio rerio (NM_001242995.1) was prepared to generate kdm1a modified (m) mRNA and was assembled from synthetic oligonucleotides. Modifications in the triplet code ($n = 7$ silent mutations) were inserted in the sequence corresponding to the mRNA AUG translational start site (i.e., Mo binding site) to prevent Mo recognition in rescue experiments. The fragment was inserted in the pMA (GeneArt, Invitrogen, Carlsbad, CA) cloning vector and cloned in transformed Escherichia coli bacteria (strain K12/DH10B, Invitrogen, Carlsbad, CA) and then purified. The final construct was verified by sequencing and the sequence identity within the insertion site was 100%.

**Site-directed mutagenesis.** Site-directed mutagenesis of Kdm1a was carried out with Q5 Site-Directed Mutagenesis kit (New England Biolabs, Ipswich, MA, cat. E0554) according to manufacturers' instructions. Standard primers for kdm1a were used for exponential amplification of the plasmid DNA (F 5′-GACAGCCAGTCG AGGAGAAC-3′ and R 5′-TGCGACGTACGAGTATGAGC-3′), and mutagenic primers to create substitution of Cys334-to-Ala (C334A) in the plasmid were designed with the software NEBase Changer (F 5′-AAAACAGAAGGCTCCCC TCTATGA GGC-3′ and R 5′-ATCTTAGCCAGCTCCATATTG-3′).

**In vitro transcription of kdm1a.** Wild type and mutated kdm1a mRNA (C334A), with 7-methyl guanosine cap structure at the 5′end and poly(A) tail at the 3′end, was transcribed from the constructs using HiScribe T7 ARCA mRNA Kit (New England Biolabs, Ipswich, MA, cat. E2060) following manufacturers' instructions.

**Rescue of kdm1a knockdown by Kdm1a mRNA.** To determine whether the effects of the kdm1a KD on zebrafish embryos phenotype and tailfin regeneration were specifically due to loss of kdm1a, we co-injected kdm1a-Mo with kdm1a mRNA wild-type as a rescue. A bolus of 1 nL of solution containing 0.5 ng of kdm1a-Mo a a and 1 ng of Kdm1a RNA wild-type was injected into each egg.

**Rescue of kdm1a knockdown by kdm1a mRNA C334A.** Co-injection of kdm1a -Mo and kdm1a mRNA C334A was performed to assess whether the absence of the S-nitrosylated cysteine affected the ability of the mRNA to rescue phenotype and tailfin regeneration associated with kdm1a KD. A bolus of 1 nL of solution containing 0.5 ng of kdm1a-Mo and 1 ng of kdm1a mRNA C334A was injected into each egg.

**Pharmacological modulation of S-nitrosylation.** Adult zebrafish were anesthetized in tricaine 0.05 mg/mL. A solution of 2 μL of iNos inhibitor N(ω)-nitro-L-arginine methyl ester (L-NAME, Sigma-Aldrich, St Louis, MO, cat. N5751) 10 or 50 mM diluted in sterile PBS (from stock solution of 250 mM), or of nitric oxide (NO) scavenger Phenyl-4,4,5,5-tetramethyl imidazoline-1-oxyl 3-oxide (PTIO, Sigma-Aldrich, St Louis, MO, cat. P5084) 3 or 10 mM diluted in sterile PBS (from stock solution of 100 mM), or of NO donor S-Nitroso-N-acetyl-DL-penicillamine (SNAP, Sigma-Aldrich, St Louis, MO, cat. N3398) 10 or 30 mM diluted in DMSO (from stock solution of 100 mM), or PBS as control, was loaded on a glass capillary, prepared in advance with a micropipette puller (Narishige, Inc., PC-10) and connected to a microinjector (IM300 Microinjector; Narishige, Tokyo, Japan) and was injected into the retro-orbital vein as previously described[41] on days 6, 4, 2, and 0 before tailfin amputation.

The concentration of the Nos inhibitors, L-NAME and 1400W, were adopted after a pilot study with doses up to 250 mM. The survival was recorded and fish were monitored for any physical or behavioral abnormalities at a range of doses. For both compounds at a dose of 250 nM, survival after four injections was approximately 40%, and lethargy and reduced swim were observed. At 150 mM the survival increased to 65% with no evident abnormal behavior; whereas at 50 mM, the dose adopted for the study, the survival was 85% and no evident defects were observed (Supplementary Fig. S2C).

**Optimization of the injection procedure.** In our pilot studies, injection of physiological solutions every other day and alternating the eye at each injection reduced fish mortality. In this way, the effect of a drug on fish survival/mortality and phenotype can be better evaluated. Therefore, all the solutions of drugs were injected in the retro-orbital veins every other day and alternating the eyes, so that each eye was injected only twice at a distance of four days.

**Defining the zebrafish phenotype.** Whole embryo phenotype following Mo and mRNA treatments were described on the basis of the following morphologic features observed under bright-field microscopy: reduced body length, curved body, reduced swimming, chorionated larvae at 4 dpf, edema. The phenotype was assessed using a simple six points scoring system, according to the severity of that feature and where one point was normal. At least four different clutches of larvae were assessed under each of the treatment groups. Data were reported graphically as divided in two groups: normal, i.e., embryos not showing any abnormal features, and abnormal, i.e., embryos showing one or more of the features described above.

**Cardinal vein blood velocity.** Blood velocity was estimated in the posterior cardinal vein[42] by assessing frame by frame motion of single blood cells determined from video images captured in the zebrafish tail at the level of the cloaca. Four erythrocytes per fish (at least five embryos per group) over ten frames at video frame-rate of 30 frames per second (fps) were analyzed using ImageJ to determine mean blood cell velocity ($\mu m \, s^{-1}$).

**Kaplan–Meier analysis of survival.** Kaplan–Meier analysis was used to measure the survival of adult zebrafish or larvae following each defined treatment, using PRISM 7 software.

**Immunoprecipitation.** Immunoprecipitation experiments were performed using the Pierce Classic Magnetic IP/Co-IP Kit (Thermo Fisher Scientific, San Jose, CA, cat. MAN0011737), according to manufacturer's instructions.

**RNA extraction and quantitative PCR.** mRNA was extracted from embryos using column purification (RNeasy Mini Kit, Qiagen, Hilden, Germany, cat. 74104) according to the manufacturer's instructions. Working surfaces were cleaned with RNase Zap (Thermo Fisher Scientific, San Jose, CA, cat. AM9780) to deactivate environmental RNase. Efficient disruption and homogenization of tissue was done using sterile RNase-free disposable pestles (Thermo Fisher Scientific, San Jose, CA, cat. 12-141-368) mounted on a cordless motor for 30 s and then passing the lysate 5–10 times through the needle (18–21 gauge) mounted on a RNase free syringe. RNA integrity was assessed on basis of 18S and 28S ribosomal RNA (rRNA) bands. mRNA was reverse transcribed into cDNA using qScript cDNA Synthesis Kit (Quanta Bio, Beverly, MA, cat. 95047), Primers (IDT Technologies, Coralville,

Iowa) targeting genes of interest (see Table S2) and SYBR Green PCR kit (Invitrogen, Carlsbad, CA) were used for real-time qPCR, that was performed with the QuantStudio 12 k Flex system (Applied Biosystems, Foster City, CA) following the manufacturer's instructions. Gene expression was expressed as relative fold changes using the ΔCt method of analysis and normalized to β-actin.

**Western blotting**. Lysates containing 20 μg of protein each were added with Laemmli buffer (4×) and deionised water to reach a final volume of 20 μL. A sample containing pre-stained protein standard (BioRad, Hertfordshire, UK, cat. 1610375) was used to assess molecular mass of protein bands. Samples were heated at 95 °C for 5 min and loaded on a polyacrylamide gel (4–15% gradient) (BioRad, Hertfordshire, UK, cat. 4561083). Electrophoresis was performed for 30 min at a voltage of 100 V and then for 60 min at 150 V. Gels were transferred on PVDF membranes (Amersham Hybond, Sigma-Aldrich, St Louis, MO, cat. GE10600023) for 2 h at 100V. Membranes were blocked with non-fat milk 5% in PBST (PBS +0.1% Tween) for 1 h at RT and probed with primary antibody overnight at 4 °C. Antibodies used were: Kdm1a rabbit polyclonal (1:200, Thermo Fisher Scientific, San Jose, CA, cat. PA1-41697); anti-iNos mouse monoclonal (1:200, BD Transduction Laboratories, San Jose, CA, cat. 610432); anti-β-tubulin rabbit polyclonal (1:500, Abcam, Cambridge, UK, cat. ab6046), Anti-Histone H3 nuclear marker, rabbit polyclonal (1:500, Abcam, Cambridge, UK, cat. ab1791); anti-Histone H3 (1:500, unmodified Lys4), mouse monoclonal (1:500, Merck Millipore, Massachusetts, USA, cat. 05-1341); anti-monomethyl-Histone H3 (Lys4), rabbit polyclonal (1:500, Merck Millipore, Massachusetts, USA, cat. 07-436); anti-dimethyl-Histone H3 (Lys4), rabbit monoclonal (1:500, Merck Millipore, Massachusetts, USA, cat. 04-790); anti-monomethyl-Histone H3 (Lys9), rabbit polyclonal (1:500, Merck Millipore, Massachusetts, USA, cat. ABE101); anti-dimethyl-Histone H3 (Lys9) rabbit polyclonal (1:500, Merck Millipore, Massachusetts, USA, cat. 07-212); anti-Rcor1 rabbit polyclonal (1:200, Invitrogen, Carlsbad, CA, cat. PA5-41564); anti-Hdac1 rabbit polyclonal (1:200, Abcam, Cambridge, UK, cat. ab33278); anti-Rbbp4 rabbit polyclonal (1:200, Biorbyt, cat. orb583248); anti-Chd4 rabbit polyclonal (1:200, Biorbyt, Cambridge, UK, cat. orb575051); anti-TMT mouse monoclonal (1:200, Thermo Fisher Scientific, San Jose, CA, cat. 90075). Membranes were washed three times (5 min per wash) with PBS and incubated with HRP-conjugated goat anti-mouse (1:2000, Santa Cruz Biotechnology, Dallas, USA, SC-2005) or anti-rabbit (1:2000, Santa Cruz Biotechnology, Dallas, USA, SC-2004) antibodies for 1 h at RT. Then, membranes were washed again three times with PBS (5 min per wash). Antigen-antibody complexes were detected by incubation for 5 min to the enhanced chemiluminescence solution (ECL, Amersham) followed by exposure to a photographic film (BioMax XAR Film Kodak, Sigma-Aldrich, St Louis, MO). The film was developed and band density was quantified by densitometry using ImageJ. β-tubulin and Histone H3 were used as loading control for cytoplasmic and nuclear protein, respectively.

**Sequence and structural analysis of Kdm1a protein**. Similarities of zebrafish and human Kdm1a proteins were assessed using Protein Blast (https://blast.ncbi.nlm.nih.gov/Blast.cgi). Kdm1a crystal structure was obtained from the Protein Data bank (https://www.ebi.ac.uk/pdbe/entry/pdb/6nqm).

**Enzymatic isolation of endothelial cells from zebrafish tailfin**. Cells were isolated according to[43] with some modifications. In brief, amputated tailfin from adult Tg(fli1:EGFP)[y1] zebrafish were placed in chilled PBS, washed with calcium-free Ringer solution (116 mM NaCl, 2.6 mM KCl, and 5 mM Hepes, pH 7.0), and replaced with 1 mL solution of trypsin 0.25% (Gibco) added with 50 μg collagenase P (Roche) and 1 mM EDTA. Tissue was disaggregated first using fine scissors and then by pipetting the solution with a 200 μL pipette tip every 5 min for about 30 min. Cell suspensions were filtered through a 40-μm cell strainer (BD Biosciences) into FACS tubes.

**Flow cytometry characterization of fli1+ cells from zebrafish**. Cell samples were run on a BD FACS Aria (BD Biosciences). FSC-H and FSC-A were used to select cell singlets; 4′,6-diamidino-2-phenylindole (DAPI) to select viable single cells; wild-type (Wik) zebrafish were used to set the gate between GFP− (i.e., fli1−) and GFP+ (i.e., fli1+) cells. At least 10,000 of fli1− and fli1+ cells (excitation [Ex]: 488 nm; emission [Em]: 530 nm) were sorted into chilled PBS and 10% fetal bovine serum for further analysis. FlowJo 10 (Becton and Dickinson) was used to analyse data.

**Statistical analysis**. Results were expressed as the mean ± SEM. Each experiment was performed three times (biological replicates). The Shapiro–Wilk test was used to confirm the null hypothesis that the data follow a normal distribution. Statistical comparisons between two groups or multiple groups were then performed, respectively, via Student t-test or ANOVA test using PRISM 7 software followed by Bonferroni post hoc test. Log-rank test and Gehan–Breslow–Wilcoxon test were used for statistical analysis of the Kaplan–Meier curves. A p value < 0.05 was considered significant.

**Reporting summary**. Further information on research design is available in the Nature Research Reporting Summary linked to this article.

## Data availability

TMT-labeled S-nitrosylated protein analysis mass spectrometry data have been deposited to the ProteomeXchange Consortium via the MASSIVE repository (MSV000085055) with the dataset identifier PXD17883 and are freely available. Furthermore, a full list of the S-nitrosylated proteins derived from the mass-spec is included in this manuscript as Dataset S1. Gene Expression Omnibus (GEO) database, accession GSE137971 was used for single cell sequencing analysis. All other relevant data supporting the key findings of this study are available within the article supplementary files, and Source Data file. Source data are provided with this paper.

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

## Acknowledgements

This project has received funding from the European Union's Horizon 2020 research and innovation programme under the Marie Sklodowska-Curie grant agreement n. 797304 (to G.M.). This work was also supported by the British Heart Foundation (BHF) CoRE Award (RE/13/3/30183) and Transition Fellowship (RE/18/5/34216) (to G.M.); the BHF Chair of Translational Cardiovascular Sciences (CH/11/2/28733) and Centre for Regenerative Medicine (RM/17/3/33381) (to A.H.B.); the Cullen Trust for Health Care (to J.P.C. and G.M.); and the National Institutes of Health R01 Grants HL 148338 and HL133254 (to J.P.C.). Furthermore, we are grateful to David E. Newby, supported by the British Heart Foundation awards CH/09/002, RG/16/10/32375, RE/18/5/34216) and the Wellcome Trust award WT103782AIA, for providing additional funding support. We thank John F. Rawls lab (Duke University School of Medicine) for providing *Tg(nfkb:EGFP)^nc1* zebrafish.

## Author contributions

G.M. is the senior author of this work and is primarily responsible for the conception, design and experimental investigation, data collection and analysis, resources, and original and revised drafts; S.Y.J., J.M.C., A.J., and H.E.L. contributed to mass spectrometry run and dataset collection; C.C., K.R., and J.R. contributed to bioinformatic analysis; M.A.D. and A.H.B. contributed resources, advised on experimental design and contributed to manuscript editing and discussions; J.P.C. contributed to conceptualization, resources and manuscript editing.

## Competing interests

The authors declare no competing interests.
