## [Peer Review File · Nature Communications]

Nuclear S-nitrosylation impacts tissue regeneration in zebrafish

Editorial Note: Parts of this Peer Review File have been redacted as indicated to maintain the confidentiality of unpublished data. When text is deleted in rebuttals and referee reports, add “[redacted]” in that location.REVIEWER COMMENTS

Reviewer #1 (Remarks to the Author):

Zebrafish tailfin regeneration is a classic model. Inducible nitric oxide synthase (iNOS) is involved in tissue regeneration, but its importance is not generally appreciated. It is known that iNOS is capable of nuclear translocation to facilitate S-nitrosylation of nuclear proteins. In the manuscript “Nuclear S-nitrosylation impacts tissue regeneration”, the authors propose a novel pathway by which tissue injury induces nuclear translocation of iNOS. Nuclear iNOS activity results in temporal S-nitrosylation of KDM1a and negatively regulates its binding to the CoRest complex—ultimately resulting in impaired demethylation of H3K4 and facilitation of tissue regeneration. In general, these findings are novel (both from standpoint of tissue regeneration and PTM crosstalk) and of interdisciplinary interest. However, the following concerns should be addressed prior to publication:

Major points:

L-NAME is a nonspecific inhibitor of nitric oxide synthase. The data in Fig 1A suggests that both NOS1 and NOS2b are upregulated after injury. For these reasons it is unclear whether NOS1 is contributing to the regeneration phenotype observed after L-NAME and PTIO administration (Fig 1F). The authors should consider examining tail fin regeneration in the context of NOS2b knock-down or with a specific iNOS inhibitor.

It is unclear whether the data depicted by the blue line in Fig 3B is derived from LC-MS/MS analysis or western blotting. In either case, the authors should consider confirming the enhancement of KDM1a S-nitrosylation after amputation by western blotting and including this in the figure.

The authors demonstrate by mass spec analysis that Cys334 is likely the primary SNO-site of Kdm1a; however, this SNO-site should be confirmed through mutation and western blot (although I would agree that the functional data with Cys mutant ultimately prove the point). Furthermore, the authors should consider including the Kdm1a SNO-site mutant in Figures 3C-G to further demonstrate the reliance of both histone methylation (3C-D) and CoRest complex association (3F) on S-nitrosylation at this cysteine.

Minor points:

The authors should include total protein loading controls for the experiments in Figures 1D and 1E.

The authors should clarify the total number of repeats used for each condition for the LC-MS/MS data presented in Figure 2.

It is unclear whether the 'Kdm1a KD' group represented in Figures 3C and 3D was performed on uninjured or injured tissue. The authors should clarify this.

The fonts should be the same in every figure

Replace "mut" with "C334A" in figures.

--

Reviewer #2 (Remarks to the Author):

In the manuscript entitled "Nuclear S-nitrosylation impacts tissue regeneration", Gianfranco and colleagues demonstrated that Kdm1a is S-nitrosylated through the nitric oxide (NO) signaling upon tailfin amputation to facilitate fin regeneration. qPCR and western blot analysis revealed that the nuclear level of Nos2, presumably Nos2b, is transiently increased during fin regeneration, implicating the roles of S-nitrosylation in nuclear proteins during regeneration. To identify S-nitrosylated nuclear proteins, the authors labeled S-nitrosylated proteins using the iodoTMT method and subsequently conducted LC/MS/MS. This screening found Kdm1a/Lsd1 as a potential S-nitrosylated nuclear protein that may influence fin regeneration. S-nitrosylation on Cys334 is likely important for physical interaction between Kdm1a/Lsd1 and CoRest complex. Knocking down Kdm1a through morpholino (MO) treatment suggests that Kdm1a/Lsd1 plays crucial roles in fin regeneration.

The idea of this report is promising. Identifying new regeneration-associated factors using proteomics screening, such as iodoTMT, is very impressive, providing an innovative methodology for regenerative biology. S-nitrosylation of Kdm1a and its roles in fin regeneration also contribute to developing a new concept connecting injury, chromatin modification, and regeneration. Despite these strengths, this manuscript is somehow descriptive without providing mechanistic roles of S-nitrosylation on fin regeneration. Thus, the author should address the following concerns:

1. Cellular or molecular mechanisms affected by S-nitrosylation and/or Kdm1a/Lsd1.

Appendage/fin regeneration undergoes three prominent phases: 1) wound/regeneration epidermis formation, 2) blastema formation and 3) regenerative outgrowth and patterning. Upon amputation, the epidermis forms a special cell layer, referred to as the wound/regeneration epidermis. This wound/regeneration epidermis expresses various signaling factors to help form blastema, a key structure for appendage regeneration, in the mesenchymal cell population. Cells in blastema proliferate and contribute to regrowth. Cells that exit from the blastema start to differentiate for functional restoration. In regenerating fins, blastema is located at the distal region while proximal cells differentiate. Although the authors identified that Kdm1a/Lsd1 is highly nitrosylated at 5 dpa, it is unclear which cell populations contain nitrosylated Kdm1a/Lsd1. Is Kdm1a/Lsd1 highly enriched in the blastema or cells of the differentiating zone? If blastema, what are the roles of kdm1a/Lsd1? Does kdm1a/Lsd1 mediate repression of specific genes? If other cell-types rather than blastema, how does kdm1a/Lsd1 impact fin regeneration? These data should be addressed to provide mechanistic insights into the roles of S-nitrosylated Kdm1a/Lsd1 in fin regeneration. The authors must address these by performing further experiments.

a) nos1, nos2a, nos2b, kdm1a/Lsd1, rcor, and rbbp4 spatiotemporal expression pattern

in situ hybridization of immunostaining must be performed to determine which cells express these genes. In particular, nos2b immunostaining data would be required to support their biochemical results as well as to identify cell-types.

b) H3K4me1, H3K4me2, H3K4me3 ChIP-seq and RNA-seq analysis

These profiles will be required to identify kdm1a/Lsd1 target regions and genes.

c) Which cellular mechanisms are influenced by kdm1a/Lsd1?

Fin is complex tissues containing many distinct cell-types, including epidermis, fibroblast/mesenchyme, osteoblast, endothelial cells, and so on. The cellular mechanisms affected by kdm1a/Lsd1 as well as S-nitrosylated proteins are unclear. The authors should examine proliferation, osteogenesis, and/or vasculogenesis.

2. Kdm1a/Lsd1 morpholino (MO) experiments

Although the authors validate their MO in Fig 4D, the MO may show off-target effects. For the adult MO experiment, the authors employed vivo-MO but the efficiency was unlikely tested. I assume that Fig 4D was done with embryos rather than adult fins. It is also unclear whether mRNA injected with vivo-MO is successfully delivered to the fin tissues. In addition, these MO experiments could cause systemic kdm1a/Lsd1 knockdown, suggesting that impaired fin regeneration may be a secondary effect.

For example, a previous study demonstrated that lsd1 mutants die by 10 dpf due to impaired hematopoiesis. This data raises the question that immune cells are not functional, causing fin

regeneration defects. Thus, the authors should employ more targeted approaches to control *kdm1a/lzd1* function, such as CRISPR/cas9-mediated loss-of-function or transgenesis-mediated overexpression studies.

3. The relationship between NFkB and iNOS expression.

It is unclear whether the NFkB activation drives NOS upregulation. Functional manipulation of NFkB is required. If not available, I recommend removing NFkB reporter expression during fin regeneration.

4. Representative fin images

Representative images are not provided for Figs 1F, 4E, S1B. Also, the quantification method used in Fig. S1B is not mentioned. There is no description of how to measure “% tailfin regrowth” in the method section.

--

Reviewer #3 (Remarks to the Author):

The question of how epigenetic and signaling processes are integrated to promote tissue regeneration is the central question explored in this manuscript by Matrone et al. The premise is that nitric oxide (NO) signaling via inflammatory processes induced by tissue injury (amputation in this case) changes epigenetic modifiers to make the chromatin more accessible to pro-regenerative factors. It is implicit that the outcome would be a change in gene expression. The hypothesis that inflammatory signaling after amputation could cause global changes in post-translational modifications of epigenetic modifiers favoring chromatin and cellular plasticity during regeneration is of high interest, which makes this study timely and novel. The model system chosen is a good one: zebrafish tail amputation is a well established model for regeneration, and the investigative team identify the proteins that are modified by S-nitrosylation and then to investigate the role of one of these proteins, the histone lysine demethylase, KDM1a.

The approach is sound and the experiments showing that pharmacological inhibition of iNOS reduces regeneration as does genetic depletion of KDM1a are compelling. The data are clear; a significant finding of this work is that nuclear proteins were S-nitrosylated throughout different time-points of regeneration and that the majority of the proteins were modified uniquely at a specific time-point, and that reducing KDM1a blocks regeneration. However, the major questions remain to be further explored to complete this study as, in the current stage, it appears somewhat preliminary. The major one is

showing that it is the S-nitrosylation of KDM1a that reduces its activity in the context of regeneration by changing gene expression. Since KDM1a can both repress and activate genes depending on the context, it is important to close the loop here by determining what effect S-nitrosylation has on KDM1a activity towards both H3K4me3/2/1 – as shown in Figure 3 – and towards HeK9me3, which is not explored.

Major

1. The study lacks sufficient data to support the proposition that S-nitrosylation of KDM1a is essential for tailfin regeneration. If S-nitrosylation blocks the functionality of KDM1a as normal response during regeneration, it is not clear how the ablation of whole protein that should lead to a similar result, removing the enzymatic activity of KDM1a, it is actually impairing regeneration. Further, how the mutation of nitrosylation site C334A that should lead to a constitutively active KDM1a have the same phenotype of the knockdown.
2. Does blocking S-nitrosylation lead to distinct changes in gene expression regulated by KDM1a?
3. What is the purpose of the *kdm1a* mutant shown in Figure 4B? Or the morpholino knockdown in embryos shown in Fig. 4A?
4. The technical approach for KDM1a depletion in the tail of zebrafish adults is not clear. How does injection of morpholino or RNA into the circulation sustains KDM1a knockdown in the tail – or rescues it when the RNA is replaced? The results are profound, but the technical basis of this is not clear. Why was the mutant that was generated not used instead?
5. Analysis of the epigenetic effects of S-nitrosylation on KDM1a and regeneration is incomplete: how does this effect gene activation? Suppression? Which pathways?
6. Does the tail regenerate when iNos inhibitors are removed?
7. Further analysis of the blunted regeneration phenotype is warranted. What cells is KDM1a and iNos acting on during tissue regeneration? The blastema? The processes leading to angiogenesis (as suggested by the pathway analysis)? Does it promote proliferation?

Minor

1. Data from Figure S5 would be beneficial to include in the main figure.
2. In Fig1B it is not clear whether the levels of protein NOS2 in the total extract changes over time, since gene expression in Fig1A suggests a mild significant increase. Adding a lane showing total protein extract on the western blot would resolve this.
3. In many of the graphs, although error bars are shown, it is not clear whether these are SE, SD and, it is remarkable in some cases how very tight the data is. It would be better to plot as a display of all the data points so that the reader can see the range.

4. Many other S-nitrosylated proteins were identified in this screen. It is remarkable that KDM1a is the rate limiting protein. More explanation of the other proteins identified and how they can interact with KDM1a would benefit the study.
5. Discrepancies between text and M&M on how pharmacological modulation of S-nitrosylation was achieved need to be fixed. It is not clear if iNOS inhibitor and/or NO scavenger were administered in the water or injected into the retro-orbital vein.
6. The authors describe pharmacological modulation of S-nitrosylation on days 6, 4, 2 and 0 before tailfin amputation, but it is not clear if this is the same treatment protocol used for data presented in Fig1D-F. Regardless of how the iNos inhibition was delivered, it is not clear how the authors control for the effects of iNos inhibition in whole fish for a week, as it would be predicted that this global NO inhibition would affect multiple aspects of the fish physiology and viability. This should be added in supplemental information.
7. In Fig1A it is not clear what the y-axis label represents, i.e. fold change or ddCt. Is this degree of induction of NOS2 sufficient to trigger the subsequent nuclear protein S-nitrosylation or is the effect on nuclear proteins only attributed to its nuclear translocation?
8. In Fig1E, it is not clear if PBS it has been used in both the first 2 lane of controls. If yes, it is not clear why there are differences between lane 1 and 2.
9. In Fig1F, it would be helpful to have representative images of this interesting observed regeneration defect.

We thank the reviewers for their very constructive comments. We have made the necessary
changes, including expansive new experimental work to the manuscript according to their
recommendations. We believe that the manuscript has been substantially improved based on
their insightful comments.

Reviewer #1 (Remarks to the Author):

Zebrafish tailfin regeneration is a classic model. Inducible nitric oxide synthase (iNOS) is
involved in tissue regeneration, but its importance is not generally appreciated. It is known
that iNOS is capable of nuclear translocation to facilitate S-nitrosylation of nuclear proteins.
In the manuscript “Nuclear S-nitrosylation impacts tissue regeneration”, the authors propose
a novel pathway by which tissue injury induces nuclear translocation of iNOS. Nuclear iNOS
activity results in temporal S-nitrosylation of KDM1a and negatively regulates its binding to
the CoRest complex—ultimately resulting in impaired demethylation of H3K4 and
facilitation of tissue regeneration. In general, these findings are novel (both from standpoint
of tissue regeneration and PTM crosstalk) and of interdisciplinary interest. However, the
following concerns should be addressed prior to publication:

*We thank the reviewer for highlighting the novelty and interdisciplinary interest associated*
*with our manuscript.*

Major points:

L-NAME is a nonspecific inhibitor of nitric oxide synthase. The data in Fig 1A suggests that
both NOS1 and NOS2b are upregulated after injury. For these reasons it is unclear whether
NOS1 is contributing to the regeneration phenotype observed after L-NAME and PTIO
administration (Fig 1F). The authors should consider examining tail fin regeneration in the
context of NOS2b knock-down or with a specific iNOS inhibitor.

*Thank you for this comment. To address this, we have now included new experiments using*
*N-(3-(aminomethyl)benzyl)acetamide, also called 1400W, a tight binding, potent and*
*highly selective inhibitor of inducible nitric oxide synthase ($K_i = 7 \text{ nM}$) (Garvey E et al.,*
*JBC, 1997). It is more selective over nNOS which K_i values is $2 \mu\text{M}$. Treatment of zebrafish*
*with 1400W 50 mM significantly reduced tailfin regeneration as similar as L-NAME,*
*suggesting that NOS2 (iNOS) is critical for tailfin regeneration, see lines 110-114 of the*
*Results section and lines 815-821 in methods section, and Fig. S2A.*

It is unclear whether the data depicted by the blue line in Fig 3B is derived from LC-MS/MS
analysis or western blotting. In either case, the authors should consider confirming the
enhancement of KDM1a S-nitrosylation after amputation by western blotting and including
this in the figure.

*Apologies for the confusion. Indeed, the blue line in Figure 3B is derived from mass-spec*
*analysis, as now mentioned in the main text at page 7, line 149-150. To confirm this MS data,*
*we have now included in Figure 3C the western blotting showing the enhancement of KDM1a*
*S-nitrosylation after amputation, lines 151-152 of the main text and lines 668-679 and 865-*
*896 in the methods. For these experiments, proteins extracted from the regenerating and*
*control tailfins were labelled with TMT, then were immunoprecipitated with KDM1a*
*antibody and finally blotted using the antibody detecting TMT. This western blotting data*
*strengthens the mass-spec analysis, showing the S-nitrosylation of KDM1a at 3 and 5 days*
*post-amputation whereas this S-nitrosylation is absent in the other time point (10 days).*

The authors demonstrate by mass spec analysis that Cys334 is likely the primary SNO-site of
Kdm1a; however, this SNO-site should be confirmed through mutation and western blot
(although I would agree that the functional data with Cys mutant ultimately prove the point).
Furthermore, the authors should consider including the Kdm1a SNO-site mutant in Figures
3C-G to further demonstrate the reliance of both histone methylation (3C-D) and CoRest
complex association (3F) on S-nitrosylation at this cysteine.

*Thank you for this comment. In the previous version of the manuscript, we included the data*
*of SNO-site mutation in figure 4 (now **figure 6**). In fact, in support of figure 3 and to confirm*
*the role of the S-nitrosylation on Cys334, we performed site-directed mutagenesis for*
*KDM1a, where Cys334 of KDM1a mRNA was replaced with Alanin (C334A), Fig. 6A. This*
*mutated (C334A) version of KDM1a mRNA was used to assess the effects of the absence of*
*the KDM1a C334 S-nitrosylation site on the zebrafish phenotype and on tailfin regeneration*
*after KDM1a knockdown. To assess the specificity of the morpholino for the gene of interest,*
*a rescue is normally performed by co-injecting the morpholino with the mRNA for the gene of*
*interest (an mRNA form that is not recognised by the morpholino either by removing the*
*appropriate regions or by creating silent mutations in the morpholino binding site). In our*
*experiments, the mutated (C334A) or the wild-type version of KDM1a mRNA were co-*
*injected with KDM1a morpholino in zebrafish embryos. While the co-injection with the wild-*
*type mRNA was able to rescue all the morpholino defects, the co-injection of the mutated*
*mRNA with the morpholino was unable to rescue the phenotypic effects of the morpholino on*
*tailfin regeneration (Fig 6B-D). This demonstrated that the S-nitrosylation of KDM1a is*
*required for tissue regeneration. The text is included in the methods section lines 773-806*
*and the results section lines 251-284.*

Minor points:

The authors should include total protein loading controls for the experiments in Figures 1D
and 1E.

*We have now added Histone H3 as loading controls in figures 1D and 1E.*

The authors should clarify the total number of repeats used for each condition for the LC-
MS/MS data presented in Figure 2.

*We have now added this information in the legend of figure 2, reporting that LC-MS/MS data*
*was performed twice (N=2). Both repeats showed similar changes in S-nitrosylated proteins*
*during regeneration and, in particular, of KDM1a.*

It is unclear whether the 'Kdm1a KD' group represented in Figures 3C and 3D was
performed on uninjured or injured tissue. The authors should clarify this.

*We have now reported in figure 3C-D (now **Fig. 4D-E**) and in the respective legend that*
*KDM1a KD western blotting was performed in injured tailfin.*

The fonts should be the same in every figure

*We have corrected these mistakes.*

Replace "mut" with "C334A" in figures.

*We have now fixed this and apologise for the oversight.*

Reviewer #2 (Remarks to the Author):

In the manuscript entitled “Nuclear S-nitrosylation impacts tissue regeneration”, Gianfranco
and colleagues demonstrated that Kdm1a is S-nitrosylated through the nitric oxide (NO)
signaling upon tailfin amputation to facilitate fin regeneration. qPCR and western blot
analysis revealed that the nuclear level of Nos2, presumably Nos2b, is transiently increased
during fin regeneration, implicating the roles of S-nitrosylation in nuclear proteins during
regeneration. To identify S-nitrosylated nuclear proteins, the authors labeled S-nitrosylated
proteins using the iodoTMT method and subsequently conducted LC/MS/MS. This screening
found Kdm1a/Lsd1 as a potential S-nitrosylated nuclear protein that may influence fin
regeneration. S-nitrosylation on Cys334 is likely important for physical interaction between
Kdm1a/Lsd1 and CoRest complex. Knocking down Kdm1a through morpholino (MO)
treatment suggests that Kdm1a/Lsd1 plays crucial roles in fin regeneration.

The idea of this report is promising. Identifying new regeneration-associated factors using
proteomics screening, such as iodoTMT, is very impressive, providing an innovative
methodology for regenerative biology. S-nitrosylation of Kdm1a and its roles in fin
regeneration also contribute to developing a new concept connecting injury, chromatin
modification, and regeneration. Despite these strengths, this manuscript is somehow
descriptive without providing mechanistic roles of S-nitrosylation on fin regeneration. Thus,
the author should address the following concerns:

*We thank the reviewer for highlighting the impressive and innovative methodology and data*
*supporting new conceptual advances. We have focused on improving the mechanistic insight.*
*We refer back to the data reported to reviewer 1 on mechanistic rescue of the tailfin*
*regeneration by the WT but not mutant Kdm1a mRNA as providing critical mechanistic*
*insight on the requirement for S-nitrosylation of this protein for its functional rescue.*

1. Cellular or molecular mechanisms affected by S-nitrosylation and/or Kdm1a/Lsd1.

Appendage/fin regeneration undergoes three prominent phases: 1) wound/regeneration
epidermis formation, 2) blastema formation and 3) regenerative outgrowth and patterning.
Upon amputation, the epidermis forms a special cell layer, referred to as the
wound/regeneration epidermis. This wound/regeneration epidermis expresses various
signaling factors to help form blastema, a key structure for appendage regeneration, in the
mesenchymal cell population. Cells in blastema proliferate and contribute to regrowth. Cells
that exit from the blastema start to differentiate for functional restoration. In regenerating
fins, blastema is located at the distal region while proximal cells differentiate. Although the
authors identified that Kdm1a/Lsd1 is highly nitrosylated at 5 dpa, it is unclear which cell
populations contain nitrosylated Kdm1a/Lsd1. Is Kdm1a/Lsd1 highly enriched in the
blastema or cells of the differentiating zone? If blastema, what are the roles of kdm1a/lsd1?
Does kdm1a/lsd1 mediate repression of specific genes? If other cell-types rather than
blastema, how does kdm1a/lsd1 impact fin regeneration? These data should be addressed to
provide mechanistic insights into the roles of S-nitrosylated Kdm1a/Lsd1 in fin regeneration.
The authors must address these by performing further experiments.

*We thank the reviewer for his/her insightful comments. We have now included further*
*experiments to address these comments.*

*First, we have analysed the expression of KDM1a gene in the uninjured tailfin and in the*
*distal (blastema) and proximal (differentiating) regions of the zebrafish regenerating tailfin.*
*KDM1a expression was significantly higher in the differentiating region compared to the*
*blastema (Fig. S7), suggesting that KDM1a is more critical in the regrowing tissue.*

Furthermore, KDM1a in the proximal region had similar expression as in the control
uninjured tailfin (**Fig. S7**), suggesting that the S-nitrosylation, as well as gene expression, is
the key mechanism regulating the enzyme activity of KDM1a during tailfin regeneration. The
text has been updated in the results section, lines 189-197.

Next, we identified the main cell type where changes in S-nitrosylation occur. The formation
of new blood vessels is a crucial process during wound healing and tissue repair, just as in
embryogenesis and growth. During tailfin regeneration, endothelial cells within the growing
blood vessel sprouts and invade the avascular area. In agreement with previous observations
(Xu C. et al, Nature Communications, 2013), we detected the formation of new blood vessels
at 2 days post-amputation (dpa), that become a dense vascular plexus starting at 3 dpa (**Fig.**
**5A**). We isolated endothelial cells (EC) from the Tg(fli1:egfp) zebrafish tailfin, where EC
fluoresce green and found an increase in the percentage of EC (GFP⁺) from 4.3±0.3 % to
6.1 ± 0.4 %, respectively in the control (i.e. uninjured) versus injured tailfin (**Fig. 5B-C**). We
FACS purified these EC and performed real time PCR for KDM1a. We found that the
expression of KDM1a was not statistically different in EC of the injured tissue compared to
EC of control (**Fig. S8A**). In response to stress or injury, EC upregulate iNOS, with
consequent NO production (de Assis MC et al., Nitric Oxide, 2002), as we also showed in Fig
S8F-H. Therefore, we investigated if S-nitrosylation, more specifically of KDM1a, occurs in
EC during tailfin repair. We observed an overall increase in S-nitrosylated proteins in the
EC of the regenerating tissue compared to EC from control uninjured tissue, whereas there
was no apparent change in S-NO proteins in GFP negative cells in cells from injured and
control tissue (**Fig. 5D**). These data clearly indicate active regulation of S-nitrosylation in
response to injury in the tailfin regeneration model in the the endothelial compartment. The
text has been updated in results section, lines 198-218.

Next, we specifically analysed S-nitrosylated KDM1a in these EC cells from tailfin control
uninjured, injured and those injured with treatment with the nitric oxide (NO) scavenger
PTIO. While the analysis of total KDM1a did not change in the three conditions, S-
nitrosylated KDM1a was detectable in the regenerating tailfin only (**Fig. 5E**). The increased
S-nitrosylation of KDM1a in EC of the regenerating tailfin was associated with an increased
vessel density (**Fig. 5F**), which increase was blocked by the NO scavenger PTIO. The text
has been updated in results section, lines 219-225.

Methylated H3K4 is associated with active transcription. We had already shown in **Fig. 4D-**
**E** that H3K4me₂, a target of KDM1a, increases during the regeneration in association with
the reduced demethylase activity of S-nitrosylated form of KDM1a. Accordingly, we reasoned
that an increased H3K4me₂ marks might be associated with proangiogenic genes. To
address this hypothesis, we performed Chromatin Immunoprecipitation using a ChIP-grade
antibody specific for H3K4me₂, followed by PCR for 10 well-known proangiogenic factors,
including kdr, vegfaa, fgf2, angpt2, tek, tie1, cdh5, cd31, mmp2 and tbx20. ChIP-PCR
showed that the occupancy of H3K4me₂ on vegfaa and tek in the endothelial cells of the
injured tailfin was significantly higher compared to the injured group treated with PTIO or
to the control uninjured group (**Fig. 5G**). Furthermore, real time PCR showed that the
expression of 7 out of 10 of these proangiogenic factors, including vegfaa and tek, was
significantly increased (**P<0.001) during the regeneration (**Fig. 5H**), whereas it returned
to control levels following treatment with PTIO. This data clearly shows the consequential
effects on proangiogenic factors of KDM1a regulation in the tailfin. The text has been
updated in the results section, lines 228-247.

*When taken together, these findings show that endothelial cells of the regenerating tissue are*
*the main cell type where changes in S-nitrosylation of proteins occur after injury. KDM1a is*
*S-nitrosylated in these endothelial cells. S-nitrosylation of KDM1a reduces KDM1a*
*demethylase activity and, as consequence, di-methylated H3K4 will accumulate in specific*
*genes. In particular, we found increased occupancy of H3K4 on vegfaa and tek2 and an*
*increase gene expression of these and other key proangiogenic factors. This does not exclude,*
*however, that other S-nitrosylated proteins among those arising from the mass-spec dataset*
*could be implicated in endothelial cells and vascular regrowth as well as in other cell types.*
*We have put these results in the results, lines 243-250, and discussion section, lines 294-300.*

a) nos1, nos2a, nos2b, kdm1a/lsd1, rcor, and rbbp4 spatiotemporal expression pattern
in situ hybridization of immunostaining must be performed to determine which cells express
these genes. In particular, nos2b immunostaining data would be required to support their
biochemical results as well as to identify cell-types.

*Thank you for this comment. We agree with this reviewer that the analysis of the expression*
*of these genes is important for the manuscript. However, the use in situ hybridization or*
*immunostaining is perhaps sub-optimal in these experiments. In fact, Kdm1a, rcor1, rbbp4,*
*hdac1 and chd4 genes are known to have low tissue and cell type specificity*
*(Source: www.proteinatlas.org), and we reasoned therefore that a more sensitive assay would*
*provide more useful information on the expression of these genes in specific cell types.*
*Accordingly, we used real time PCR to measure gene expression. We analysed the expression*
*of nos1, nos2a and nos2b genes by real time PCR in EC and non-EC (respectively GFP+ and*
*GFP- cells) isolated from Tg(fli1:EGFP) fish in the injured and uninjured tailfin to assess*
*endothelial and non-endothelial contributions to expression. While Nos1 and Nos2a did not*
*show differences between groups, Nos2b show a significant increase in regenerating EC*
*(GFP+ cells) compared to control EC and other cell types (GFP- cells) (see Fig. S8F-H). We*
*could not detect differences for all five genes between the four groups: GFP+ and GFP- cells*
*from Tg(fli1:EGFP) in the injured and uninjured fin (see Fig. S8B-E). We have discussed*
*these results in lines 200-2018 of the main text.*

b) H3K4me1, H3K4me2, H3K4me3 ChIP-seq and RNA-seq analysis

These profiles will be required to identify kdm1a/lsd1 target regions and genes.

*Thank you for these suggestions. Indeed, global ChIP-seq and RNA-seq analyses would be*
*informative, however for the purposes of this manuscript focusing on key endothelial genes,*
*we have performed chromatin immunoprecipitation using H3K4me2 antibody and performed*
*real time PCR for a group of n=10 endothelial relevant genes, as discussed above. We*
*showed that at 5 days post-amputation (dpa) H3K4me2 occupancy to the genes VEGFaa, a*
*proangiogenic factor (Shibuya M, Genes & Cancer, 2011), and Tek, that promotes vascular*
*stability (Thurston G et al., Cold Spring Harbor perspectives in medicine, 2012) is*
*significantly increased compared to the uninjured group and the injured group previously*
*injected with the NO scavenger PTIO. Furthermore, real time PCR showed that the*
*expression of Angpt1, CD31, Cdh5, Kdr, Tek and Vegfaa genes was significantly increased in*
*the tailfin at 5 dpa compared to the uninjured group and the injured group previously*
*injected with PTIO. The text has been updated in the results section, lines 228-247.*

c) Which cellular mechanisms are influenced by kdm1a/lsd1?

Fin is complex tissues containing many distinct cell-types, including epidermis,
fibroblast/mesenchyme, osteoblast, endothelial cells, and so on. The cellular mechanisms
affected by kdm1a/lsd1 as well as S-nitrosylated proteins are unclear. The authors should
examine proliferation, osteogenesis, and/or vasculogenesis.

We agree with this reviewer that a histone modifier such as *Kdm1a* could play roles in
different cell types of the regenerating fin. In our answer to comment 1 of this reviewer (lines
148-222), we have discussed about further experiments we conducted to identify the cellular
processes most affected by S-nitrosylation, in particular of *KDM1a*. In brief, we found an
increased expression of *NOS2B* in endothelial cells after injury compared to the rest of the
tailfin tissue (**Fig. S8H**) and then found that this was associated to an increase in the S-
nitrosylation of proteins in the endothelial cells of the regenerating tissue (**Fig.5D**). As *NOS2*
is increased specifically in endothelial cells, we believe that S-nitrosylation regulates cell-
autonomously these cells. Nonetheless, S-nitrosylation could also be implicated in the
regulation of other cell types, for example immune cells.

2. *Kdm1a/Lsd1* morpholino (MO) experiments

Although the authors validate their MO in Fig 4D, the MO may show off-target effects. For
the adult MO experiment, the authors employed vivo-MO but the efficiency was unlikely
tested. I assume that Fig 4D was done with embryos rather than adult fins. It is also unclear
whether mRNA injected with vivo-MO is successfully delivered to the fin tissues. In
addition, these MO experiments could cause systemic *kdm1a/lsd1* knockdown, suggesting
that impaired fin regeneration may be a secondary effect.

For example, a previous study demonstrated that *lsd1* mutants die by 10 dpf due to impaired
hematopoiesis. This data raises the question that immune cells are not functional, causing fin
regeneration defects. Thus, the authors should employ more targeted approaches to control
*kdm1a/lsd1* function, such as CRISPR/cas9-mediated loss-of-function or transgenesis-
mediated overexpression studies.

Thank you for raising these issues that help us to clarify the knockdown strategy. When using
morpholino, we routinely check for off-target effects. In fact, we quantify *p53* gene expression
when determining the optimal concentration of a morpholino (Mo). In particular, when we
developed our dose response curve for *KDM1a* Mo (0.2, 0.4, 0.8, 1.2 and 2 ng Mo per egg),
*p53* was quantified for each point. We can confirm that we did not observe an increase in
*tP53* gene expression up to 2 ng. (**Fig. S9B**). We therefore decided not to co-inject *tP53* Mo
(see results section, lines 271-273). Nonetheless, we did observe an increase in non-specific
structural abnormalities as we increased the dose of Mo injected and these were seen to
correlate inversely with the *KDM1a* western blotting signal. We routinely record the
penetrance of the observed phenotypes and for the dose of 0.4 ng this was > 70% (line 275).

The survival rate of *KDM1a* KD embryos at 120 hpf was approximately 80 % (**Fig. S9C**).
*KDM1a* KD embryos did not show gross abnormalities compared to control (**Fig. S9D**),
however we found in these embryos a reduced blood velocity and a reduced expression of
*Gata1*, a red blood cell marker, consistent with a previous study (**Fig. S9E-F**). These effects
are consistent with those observed by Takeuchi et al. (*LSD1/KDM1A* promotes hematopoietic
commitment of hemangioblasts through downregulation of *Etv2*. PNAS, 2015).

However, to further strengthen our data on the role of *KDM1a*, in this revision we have
included additional experimental data using a splice blocking antisense oligo, with a
sequence that does not overlap with the first Mo sequence (lines 264-271 in the main text and
785-787 in material and methods), according to the guidelines for Morpholino use in
zebrafish (Stainier et al., Plos Genetics, 2016). The effects on the phenotype and on tailfin
regeneration observed with this second oligo were consistent with the first one. In addition to
the injection of the two Mo separately, at the same dose (0.8 ng per egg), we also co-injected
morpholinos at ½ dose of each (0.4 ng per egg). While the phenotype was not apparent with
½ dose of each Mo injected alone, the co-injection of the two Mo produced phenotypic effect,
similar to that produced by a single morpholino at the dose of 0.8 ng per egg. The additive

effect is a strong indication that the MO effects are specific for KDM1a. Indeed, the co-
injection of KDM1a Mo + mRNA wild type was able to rescue the Mo phenotype and further
confirmed KDM1a Mo specificity.

Morpholino injection for KDM1a KD was performed also in adult zebrafish (**Fig.4**). The KD
efficiency as well as the delivery in the tailfin were assessed by western blotting and the
effects on tailfin regeneration (**Fig. 4B-C**), see lines 171-175 in results section.

We agree with this reviewer that MO experiments could cause systemic *kdm1a/lzd1*
knockdown and, given the role of the KDM1a chromatin modifier as master regulator, we
cannot exclude that the impaired fin regeneration in KD zebrafish could, at least partially,
derive from these systemic effects. For the same reasons, a model of CRISPR/cas9-mediated
KDM1a loss-of-function would not solve this important issue. However, the principal aim of
this study was to understand the role of the S-nitrosylation, of KDM1a, rather than of the
protein per se, in tissue regeneration.

[Redacted]

In summary of the above, our experiments using two morpholinos clearly show the effects of
KDM1a KD in zebrafish tailfin regeneration. These effects are specific and reproducible and
in line with international standards. The specificity of the Morpholinos were confirmed by the
rescue of the phenotypes following co-injection with KDM1a mRNA.

3. The relationship between NFkB and iNOS expression.

It is unclear whether the NFkB activation drives NOS upregulation. Functional manipulation
of NFkB is required. If not available, I recommend removing NFkB reporter expression
during fin regeneration.

The role of NFkB in the activation of iNOS has been previously described (Hatano E. et al.,
NF-kappaB stimulates inducible nitric oxide synthase to protect mouse hepatocytes from
TNF- α - and Fas-mediated apoptosis. Gastroenterology, 2011), therefore a thorough
analysis of the role of NFkB on iNOS has not been performed for this manuscript. However,
to confirm the relationship between NFkB and iNOS expression, as requested by this
reviewer, in this revision we have included new data assessing the effects of NFkB inhibition
on iNOS regulation in adult zebrafish regenerating tailfin. Adult zebrafish were injected with
the NF-kB inhibitor Bay11-7082 or vehicle prior of tailfin amputation. Fish treated with
Bay11 showed a reduced expression of NOS2 in the regenerating tailfin compared to vehicle
(lines 85-88 of the results section and **figure S1C**).

4. Representative fin images

Representative images are not provided for Figs 1F, 4E, S1B. Also, the quantification method
used in Fig. S1B is not mentioned. There is no description of how to measure “% tailfin
regrowth” in the method section.

We have now included representative images in **Fig. 1F** for the four groups analysed. Images
for figure 4E were previously reported in figure 4C (now **Fig. 6D**). Representative images for
the **Fig. S1B** have now been included below the same dot plot. We have now included in
material and methods, in section “Zebrafish tailfin amputation and regeneration”, lines 603-
613, the method used to quantify tailfin regeneration.

Reviewer #3 (Remarks to the Author):

The question of how epigenetic and signaling processes are integrated to promote tissue
regeneration is the central question explored in this manuscript by Matrone et al. The premise
is that nitric oxide (NO) signaling via inflammatory processes induced by tissue injury
(amputation in this case) changes epigenetic modifiers to make the chromatin more
accessible to pro-regenerative factors. It is implicit that the outcome would be a change in
gene expression. The hypothesis that inflammatory signaling after amputation could cause
global changes in post-translational modifications of epigenetic modifiers favoring chromatin
and cellular plasticity during regeneration is of high interest, which makes this study timely
and novel. The model system chosen is a good one: zebrafish tail amputation is a well-
established model for regeneration, and the investigative team identify the proteins that are
modified by S-nitrosylation and then to investigate the role of one of these proteins, the
histone lysine demethylase, KDM1a.

The approach is sound and the experiments showing that pharmacological inhibition of iNOS
reduces regeneration as does genetic depletion of KDM1a are compelling. The data are clear;
a significant finding of this work is that nuclear proteins were S-nitrosylated throughout
different time-points of regeneration and that the majority of the proteins were modified
uniquely at a specific time-point, and that reducing KDM1a blocks regeneration. However,
the major questions remain to be further explored to complete this study as, in the current
stage, it appears somewhat preliminary. The major one is showing that it is the S-
nitrosylation of KDM1a that reduces its activity in the context of regeneration by changing
gene expression. Since KDM1a can both repress and activate genes depending on the context,
it is important to close the loop here by determining what effect S-nitrosylation has on
KDM1a activity towards both H3K4me3/2/1 – as shown in Figure 3 – and towards
HeK9me3, which is not explored.

*We thank this reviewer for his/her comments on our choice of model and compelling results.*
*We have provided additional experimentation to reviewers 1 and 2 as well as answers to this*
*reviewer below that add significant depth and mechanistic understanding to our research.*

*Kdm1a is not able to demethylate H3K4me3 or H3K9, according to the chemical nature of*
*the amine oxidation reaction catalyzed by flavin-containing amine oxidases, that precludes*
*H3K4me3 as a substrate (Shi Y et al. Histone Demethylation Mediated by the Nuclear Amine*
*Oxidase Homolog LSD1, Cell, 2004). However, to complete the analysis of H3K4, as*
*requested by this reviewer, we have now included in Fig. 4D-E western blotting analyses for*
*H3K4me3. Furthermore, as Kdm1a substrate specificity is modulated by interacting proteins,*
*it can demethylate H3K9 only when bound to the androgen receptor (AR). We measured the*
*level of expression of AR in the fin and found that it was very low (Fig. S5), and so decided to*
*focus on H3K4. We have also discussed this in the results section, lines 157-164 and lines*
*183-188.*

Major

1. The study lacks sufficient data to support the proposition that S-nitrosylation of KDM1a is
essential for tailfin regeneration. If S-nitrosylation blocks the functionality of KDM1a as
normal response during regeneration, it is not clear how the ablation of whole protein that
should lead to a similar result, removing the enzymatic activity of KDM1a, it is actually

impairing regeneration. Further, how the mutation of nitrosylation site C334A that should
lead to a constitutively active KDM1a have the same phenotype of the knockdown.
*Thank you for this comment that allow us to clarify a key point. The injection of KDM1a-*
*targeted morpholino did not cause gross abnormalities in the whole embryos. However, we*
*reported a slower blood circulation and reduced expression of Gata1, marker of red blood*
*cells, consistent with a previous study (Takeuchi M. et al, PNAS, 2015). We have included*
*this new data in Fig. S9E-F of this revision. KDM1a-targeted morpholino also impaired*
*tailfin regeneration in adult and larval zebrafish (Figs. 4A-C and 6B-D). In the larvae, while*
*the co-injection of the morpholino with the KDM1a mRNA wild type was able to rescue blood*
*circulation and tailfin regeneration, the co-injection with the KDM1a mRNA C334A was able*
*to restore the blood circulation but not to restore the tailfin regeneration.*

[Redacted]

*The answer could lie on the fact that KDM1a is a master regulator and is essential in*
mammals, as KDM1a knockout mice are not viable after embryonic day E7.5 (Wang J. et al.
Nature,

*2007). Zebrafish KDM1a knockout seem to be able to survive in the early embryonic stages*
*because of maternal KDM1a expression, but they die at later stages (Takeuchi M. et al,*
*PNAS, 2015).*

*The reduced H3K4 demethylase activity we observed in KDM1a KD adult zebrafish shown in*
*Fig. 4D-E could explain partially the zebrafish phenotype, but other mechanisms could be*
*implicated.*

[Redacted]

2. Does blocking S-nitrosylation lead to distinct changes in gene expression regulated by
KDM1a?

*Yes. To show this, we have now included new experiments in zebrafish injected with the nitric*
*oxide scavenger PTIO (acronym of 2-Phenyl-4,4,5,5-tetramethylimidazole-1-oxyl 3-oxide)*
*(see Fig. 5G-H). We focussed on the endothelial cells in the tailfin, where most of the*
*changes in protein S-nitrosylation occurs during regeneration. Real-time PCR analysis*
*showed that endothelial and proangiogenic genes, such as Angpt1, Cd31, Cdh5, Kdr, Vegfaa*
*and Tek were upregulated in the tailfin endothelial cells at 5 dpa compared to the uninjured*
*tailfin (Fig. 5H). However, following treatment with PTIO, the expression of these genes was*
*not upregulated but instead had values similar to the uninjured tailfin. This reduced gene*
*expression in PTIO treated zebrafish was also associated with a reduced vessel density (Fig.*
*5F). Furthermore, we showed that H3K4me2 marks on Vegfaa and Tek (see ChIP-PCR data*
*in Fig. 5G) genes was increased at 5 dpa compared to the control uninjured group. Whereas*
*the treatment with PTIO reduced the occupancy of H3K4me2 on Vegfaa and Tek genes in*

injured tailfin to values similar to those observed in the control uninjured. Methylated-H3K4
is a marker of active transcription and is demethylated by KDM1a. Therefore, the reduced
activity of the S-nitrosylated KDM1a observed during the regeneration is associated with an
increased transcription of genes regulated by H3K4. We have further discussed this in
previous comments at page 4, lines 189-211.

3. What is the purpose of the kdm1a mutant shown in Figure 4B? Or the morpholino
knockdown in embryos shown in Fig. 4A?

*The purpose of the KDM1a mutant in figure 4B (Fig. 6B-D in this revision) is to assess the*
*effects of the absence of the KDM1a C334 S-nitrosylation site on the zebrafish phenotype and*
*on tailfin regeneration. In this mutant, the cysteine in position 334 was substituted with*
*alanine, that cannot be S-nitrosylated. In particular, these experiments assessed the ability of*
*this mutated KDM1a mRNA to rescue the morpholino phenotype compared to wild-type*
*mRNA. Indeed, to assess the specificity of the morpholino for the gene of interest, a rescue is*
*normally performed by co-injecting the morpholino with the mRNA for the gene of interest*
*(an mRNA form that is not recognised by the morpholino either by removing the appropriate*
*regions or by creating silent mutations in the morpholino binding site). In our experiments,*
*the mutated (C334A) or the wild-type version of KDM1a mRNA were co-injected with*
*KDM1a morpholino in zebrafish embryos. While the co-injection of the wild-type mRNA was*
*able to rescue all the morpholino defects, the co-injection of the mutated mRNA with the*
*morpholino was able to rescue only partially the phenotypic effects of the morpholino, as*
*embryos were not able to rescue the tailfin regeneration (Fig 6A-D). This demonstrated that*
*the S-nitrosylation of cysteine 334 of KDM1a is required for tissue regeneration. We have*
*further discussed this in previous comments at page 2, lines 67-83, and page 6, lines 288-*
*337.*

4. The technical approach for KDM1a depletion in the tail of zebrafish adults is not clear.
How does injection of morpholino or RNA into the circulation sustains KDM1a knockdown
in the tail – or rescues it when the RNA is replaced? The results are profound, but the
technical basis of this is not clear. Why was the mutant that was generated not used instead?
*KDM1a KD in adult zebrafish was achieved by injection of KDM1a vivo-morpholino*
*systemically through the retro-orbital vein. In a vivo-Morpholino, the Morpholino oligo is*
*combined with an octa-guanidine dendrimer that works as a delivery moiety, that provide*
*stability to the molecule and facilitates the diffusion within the tissues. Intravenous injection*
*provides the best results for the systemic distribution of the morpholino in adult zebrafish.*
*Retro-orbital vein injection has been previously reported (Retro-orbital injection in adult*
*zebrafish. Pugach et al, JoVE, 2009) to successfully deliver a solution marked with a dye in*
*the zebrafish tailfin. This technique has been now used routinely in several labs, including*
*ours (Meng, Matrone et al, JAHA, 2016). The protocol of injection was described in the*
*material and methods, under the section “KDM1a suppression”. In brief, 2 μ L of 0.1 mmol/L*
*of vivo-Mo solution were injected every other day, alternating the left and right retro-orbital*
*vein, starting at 10 days prior tailfin amputation. The successful KDM1a knockdown in the*
*tailfin was assessed by western blotting using a KDM1a specific antibody (Fig. 4A).*
*On the other hand, mRNA is a very large molecule and it is intrinsically unstable and prone*
*to degradation by nucleases. Therefore, RNA was not injected in the adult zebrafish but only*
*in the embryos, where the cytoplasmic bridges connecting the early embryonic cells allow*
*rapid diffusion of into the cells, resulting in ubiquitous delivery (Bill BR et al., Zebrafish,*
*2009). In fact, the experimental importance of co-injecting RNA for the gene of interest is to*
*confirm the specificity of the morpholino for that gene. RNA and morpholino are co-injected*

*in the embryo at 1-2 cell stage, providing ubiquitous delivery.*

5. Analysis of the epigenetic effects of S-nitrosylation on KDM1a and regeneration is
incomplete: how does this effect gene activation? Suppression? Which pathways?

*We have performed new experiments and showed that endothelial cells of the regenerating*
*tissue are the main cell target for protein S-nitrosylation after injury and that KDM1a is*
*specifically S-nitrosylated in these cells. S-nitrosylation of KDM1a reduces KDM1a*
*demethylase activity and, as consequence, more H3K4me2 marks will accumulate in the*
*genome. In particular, we found increased occupancy of H3K4 on vegfaa and tek2 and an*
*increase gene expression of these and other key proangiogenic factors. We have discussed*
*this in previous comments, please refer page 3-4, lines 159-222, and page 5 lines 246-257.*

6. Does the tail regenerate when iNos inhibitors are removed?

*Thank you for this comment that give us the chance to make some consideration on the iNOS*
*and nitric oxide (NO) production into the vertebrates. NOS is responsible for the production*
*of most of the nitric oxide from L-arginine into the body, whereas iNOS acts mainly after*
*stress or injury following innate immune activation. However, there are also dietary and*
*environmental sources of NO, for example through the intake of nitrate and nitrite, that can*
*be oxidized to NO. In fact, an additional route in fish is the direct uptake of nitrite from the*
*ambient water to the blood across the gills. This could be one reason why zebrafish did not*
*completely lose the ability to regenerate the tailfin during the treatment with iNOS inhibitors,*
*as showed in **Fig. 1F and S2A**. Nonetheless, the initial delay in the regeneration reflects the*
*fact that at 10 days after treatment with the iNOS inhibitors, the tailfin still has not reached*
*the size of the control.*

7. Further analysis of the blunted regeneration phenotype is warranted. What cells is KDM1a
and iNos acting on during tissue regeneration? The blastema? The processes leading to
angiogenesis (as suggested by the pathway analysis)? Does it promote proliferation?

*Thank you for this comment. We have now performed further experiments showing that iNOS*
*(NOS2b in particular) and S-nitrosylated KDM1a acts more specifically on the endothelial*
*cells of the differentiating tissue, but not in the blastema, and promote neoangiogenesis. We*
*have discussed this in previous comments, please refer to page 3-4, lines 159-222, and page*
*5 lines 246-257.*

Minor

1. Data from Figure S5 would be beneficial to include in the main figure.

*Thank you for this comment. We have now embedded figure S5 in **figure 4**.*

2. In Fig1B it is not clear whether the levels of protein NOS2 in the total extract changes over
time, since gene expression in Fig1A suggests a mild significant increase. Adding a lane
showing total protein extract on the western blot would resolve this.

*We have now included the western blotting showing total NOS2 in the zebrafish regenerating*
*tailfin (**Fig. 1B**) that confirms the increase of NOS2 gene expression, detected by real time*
*PCR.*

3. In many of the graphs, although error bars are shown, it is not clear whether these are SE,
SD and, it is remarkable in some cases how very tight the data is. It would be better to plot as
a display of all the data points so that the reader can see the range.

*Error bars are expressed as Standard Error of the Mean. We have now replaced all bar*
*graphs with dot plots.*

4. Many other S-nitrosylated proteins were identified in this screen. It is remarkable that
KDM1a is the rate limiting protein. More explanation of the other proteins identified and how
they can interact with KDM1a would benefit the study.

*Undoubtedly, many interesting candidates were identified, and among them we focussed on*
*KDM1a. However, we are not sure that KDM1a is the rate limiting factor, because*
*modulating the expression and S-nitrosylation of other candidates could give similar effects*
*on the regeneration, maybe by affecting the same cells and pathways identified here but also*
*other cell types and different pathways, depending on the effects that the S-nitrosylation on*
*the molecular networking of each protein candidate. For example, HEXIM1 (hexamethylene*
*bisacetamide inducible protein 1) binds to the 7SK snRNA in the complex and inhibits the*
*CDK9 kinase activity in the P-TEFB complex (Yik et al., Mol Cell, 2003). In response to*
*different stress stimuli, P-TEFb is released from the 7SK snRNP and HEXIM and recruited to*
*the chromatin templates by the bromodomain protein BRD4 to promote transcription of many*
*cellular primary response genes (Yang et al., Mol Cell, 2005). As for KDM1a and the*
*CoREST complex, S-nitrosylation could modulate the binding of HEXIM with the members of*
*the P-TEFB complex and regulate the transcription of gene relevant during regeneration. S-*
*nitrosylation of KDM1a, HEXIM1 and maybe many other proteins could part of a molecular*
*program for a quick and reversible mechanism to switch the regulation, even before than the*
*expression, of many genes following an injury. In fact, posttranslational modifications (PTM)*
*are responsible for the protein “plasticity”. The S-nitrosylation and the crosstalk with other*
*PTMs could facilitate this plasticity. We have included this discussion, yet not exhaustive, in*
*the main text, lines 303-312.*

5. Discrepancies between text and M&M on how pharmacological modulation of S-
nitrosylation was achieved need to be fixed. It is not clear if iNOS inhibitor and/or NO
scavenger were administered in the water or injected into the retro-orbital vein.
*We have fixed this discrepancy by specifying in the main text that the drugs were injected into*
*the retro-orbital vein, line 104 of result sections.*

6. The authors describe pharmacological modulation of S-nitrosylation on days 6, 4, 2 and 0
before tailfin amputation, but it is not clear if this is the same treatment protocol used for data
presented in Fig1D-F. Regardless of how the iNos inhibition was delivered, it is not clear
how the authors control for the effects of iNos inhibition in whole fish for a week, as it would
be predicted that this global NO inhibition would affect multiple aspects of the fish
physiology and viability. This should be added in supplemental information.

*The protocol for the pharmacological modulation of the S-nitrosylation from 6 days before*
*and up to the day of tailfin amputation was described in the material and methods and results*
*reported in **Fig. 1E-F**. As shown from data of gene expression in **figure 1A**, iNOS is*
*especially activated after the injury, whereas NOS1, is more expressed in normal conditions.*
*Therefore, the inhibition of iNOS, for example by exposure to 1400W at a suitable dose,*
*should not affect other tissue significantly. Furthermore, the concentration of the NOS*
*inhibitors, L-NAME and 1400W, used in our study were adopted after a pilot study, with*
*doses up to 250mM. The survival was recorded and fish were monitored for physical or*
*behavioral abnormalities. At 250mM L-NAME or 1400W the survival after 4 injections was*
*40%, with a reduced swim. At 150mM the survival increased to 65% with no evident*
*abnormal behaviour, whereas at 50mM, the dose we adopted in the study, the survival was*
*85% and no evident defects were observed. This data has been discussed in the methods*
*section, lines 814-821, and data included in **Fig. S2B**.*

*From our experiences with injections in adult fish, we know that even 1 μ L of physiological*
*solution injected daily for a week can easily cause death, maybe for the damage that the*
*needle, even when used by expert hands, can cause in such a small animal. In our pilot*
*studies, injection of physiological solutions every other day and alternating the eye at each*
*injection reduced dramatically fish mortality. In this way, the effects of a drug on fish*
*survival/mortality and phenotype could be better evaluated.*
*Therefore, all the solutions with drug, including iNOS inhibitors, were injected in the retro-*
*twice at a distance of four days. This information is now included as a separate paragraph in*
*material and methods as “Optimization of the injection”, lines 822-829.*

7. In Fig1A it is not clear what the y-axis label represents, i.e. fold change or ddCt. Is this
degree of induction of NOS2 is sufficient to trigger the subsequent nuclear protein S-
nitrosylation or is the effect on nuclear proteins only attributed to its nuclear translocation?
*We are sorry for the misunderstanding in **Fig1A**. The label “relative expression” in the y-*
*axis in **Fig. 1A** represents 2^{dCt} , where $dCt = Ct\ NOS2 - Ct\ housekeeping\ gene$.*

*Thank you also for your second question that give us the opportunity to clarify this important*
*aspect. We believe that both NOS2 increased expression and nuclear translocation are*
*important for protein S-nitrosylation. Indeed, even a small degree of induction of NOS2 can*
*boost nitric oxide production. This because iNOS produces larger amounts of NO and for*
*prolonged periods of time (Xie and Nathan, 1994) compared to other NOS isoforms.*
*Nonetheless, NO is highly reactive molecule (being a radical with a lifetime of a few*
*seconds), and once it is converted to nitrates and nitrites by oxygen and water, cell signaling*
*is deactivated. We also hypothesise that, although NO is freely diffuses across membrane,*
*NOS2 translocation to the nuclei could help to reduce the dispersion of the NO and its*
*signalling and produce it directly into the nucleus where it will induce prtotein S-*
*nitrosylation can immediately react. In conclusion, we cannot provide a comprehensive*
*answer to the question of this reviewer, but it is highly possible that both induction of iNOS*
*gene expression and its translocation to the nucleus promote a quick and precise response,*
*that is the S-nitrosylation of hundreds of nuclear proteins.*

8. In Fig1E, it is not clear if PBS it has been used in both the first 2 lane of controls. If yes, it
is not clear why there are differences between lane 1 and 2.

*Thank you for this comment. Yes, the first two lanes in **Fig1E** are referred to fish injected*
*with PBS. However, the two lanes are biological, not technical, replicates and proteins*
*loaded in the two lanes have been extracted from different fish, therefore we expect to see*
*some variations among samples.*

9. In Fig1F, it would be helpful to have representative images of this interesting observed
regeneration defect.

*We have now included representative images in **Fig. 1F**.*

REVIEWER COMMENTS

Reviewer #1 (Remarks to the Author):

The authors have addressed my concerns and the manuscript can be published.

Reviewer #2 (Remarks to the Author):

While the authors have addressed many of my previous comments and their biochemical analysis is excellent, there are still significant major and minor concerns on zebrafish regeneration experiments.

Majors

1. kdm1a expression pattern during regeneration

The authors demonstrated that kdm1a expression is significantly higher in the differentiating region. To do this, the authors used real-time PCR (quantitative RT-PCR) for whole, distal and proximal regenerates. However, there is no detailed method describing how to collect distal and proximal regions. Indeed, collecting distal and proximal regions separately by dissection is very challenging without using microdissection or FACS sorting with transgenic fish. The authors should examine where kdm1a expresses by other methods. The potential method is in situ hybridization. kdm1a probe was described in the previous paper (Takeuchi et al., PNAS, 2015). Another possible approach is to analyze single-cell data. Several single-cell profiles are available: Hou et al., Science Adv. 2020; Wang et al., Science 2020; Alemany et al., 2018 Nature. Another approach is to do qPCR with positive control genes, such as blastema or cell cycle genes and differentiation marker genes, to validate distal and proximal regenerate samples.

2. Morpholino effect

a) Morpholino validation

Two morpholinos (MOs) were used for knock-down experiments. Although the authors described the effect of MO in Lines 265 – 270, the actual results were not provided. Moreover, it is unclear whether Fig. 6B was obtained by a single MO injection or two MOs injection. The authors mentioned “additive effects”, but it is unclear how to determine the MO effect. The appropriate way to determine the MO effect would be 1) western blot and 2) checking phenotype of kdm1a knockdown rather than abnormal development (see below). The authors should provide comprehensive results to inform how the individual or combined MO effect was determined.

The previous study (Takeuchi et al., PNAS, 2015) demonstrated that kdm1a mutation impairs hemangioblast development, leading to less blood cell formation and reduced gata1 expression. kdm1a

mutants unlikely exhibit abnormal appearances, such as edema or curved body (Fig. S2B in PNAS paper). The majority of *kdm1a* mutants can survive until 8 dpf in the reference. In the current manuscript, the authors validated morpholino (MO) effects by assessing the survival rate, appearance, *gata1* gene expression and measuring the blood velocity. First, Fig. S9 is not appropriate since it is difficult to know the phenotype of each group. The authors should change the format based on each treated sample rather than phenotypes. It is likely that regardless of MO type MO injection caused severe developmental defects, which raised questions about integrity. The authors should clarify whether normal embryos were used for further analysis or not. Second, the previous report did not test the blood velocity of *kdm1a* mutants. Please provide the rationale for why the authors measure the blood velocity. It is unclear why this analysis can represent *kdm1a* function. The more important phenotype of *kdm1a* should be hematopoietic defects and upregulation of endothelial markers (preferably tested by in situ hybridization).

b) *kdm1a* roles for the regulation of vasculogenesis gene expression

The authors treated PTIO to examine the effect of KDM1a. However, PITO treatment (Fig. 5) does not directly target KDM1a. The authors should examine whether *kdm1a* KD causes reduced expression of vasculogenesis genes and reduced enrichment of H3K4me2 (ChIP-PCR). Also, the author should provide a detailed method where they target for ChIP experiment. How to validate promoter, the length of PCR region, etc. The authors used the FINDM software to identify promoters, but it is unclear what the FINDM software is.

c) Blood vessel density

There is no detailed method for blood vessel density analysis. The authors should also provide representative images used for Fig. 5F.

d) Systemic effects

Please discuss the potential interpretation of systemic effects of *kdm1a* through immune cell regulation on fin regeneration in the discussion section.

3. S-nitrosylated nuclear proteins

The authors need to provide a more in-depth analysis of LC/MS/MS to identify S-nitrosylated nuclear proteins. The majority of proteins identified by LC/MS/MS should be nuclear proteins, but it is not demonstrated. Please describe how many percentages of identified proteins are nuclear proteins. Also, please revise dataset S1 to provide more information, such as gene names, a brief description of the function, and source (similar to Table S1).

4. Drug delivery method

First, I do recommend remove the first sentence in “the optimization of an injection procedure”: “we know that even 1 μ L of physiological solution injected daily for a week can easily cause death, maybe because of the damage that the needle, even when used by expert hands, can cause in such small animals.” This sentence may mislead researchers to consider that injecting a drug in adult fish is challenging. However, many zebrafish labs using adults routinely inject drugs without killing fish. The major reasons to kill fish by injection are likely 1) high dose of drug 2) high percentage of solvent, and 3) an injection tool. For DMSO and Ethanol, the maximum percentage would be 10-20%. The authors did

not describe how they prepare for drugs (conc. of stock, solvent to dissolve drug, etc). The authors seem to use a glass capillary but do not describe how to prepare it. The common injection tool for fish is a Hamilton syringe (1700 series having a customized gauge), not a glass capillary. Note that the cited reference 34 also used a Hamilton syringe. A glass capillary diameter may be large to significantly damage fish, leading to death. Although functional data likely support the effect of drug treatment, the authors need to clarify that doses used for regeneration study do not significantly affect mortality, like L-NAME.

Minor

1. Nomenclature: Please check the correct nomenclature for zebrafish genes, proteins, transgenic lines throughout the manuscript.

2. "Baseline" in multiple figures -> please change to uninjured throughout figures and manuscript

3. RT-qPCR

Fig 1a, S5, S7, S9. Y-axis is weird. Please revise the number. There is no information about primers. If Table S2 contains primers for qRT-PCR, please clarify it. Also, there is no primer information for b-actin.

4. Timepoint to assess regeneration

Please clarify the time point (regeneration or injured) and/or drug concentration in Fig. 1F, 3D, 3E, 3F, 4D, 4E, 5B-G, 6B-D and many supplementary figures.

5. Fig 3. Please add "IP:KDM1a" for E and F above the figures.

Reviewer #3 (Remarks to the Author):

In this revised version of the manuscript, the authors completed several new experiments that further support the conclusions that S-nitrosylation of KDM1a impairs its ability to demethylated H3K4 and this impairs fin regeneration in zebrafish. The model is elegant and several new insights are provided by the work. However, there remain a few outstanding issues that were not addressed. It would be beneficial for the authors to state outright if they are unable to address these points experimentally or do the experiments which address them directly. The new experiments showing that endothelial cells are a primary target of iNOS inhibition and KDM1a action are very interesting. Overall, the manuscript is improved and makes an interesting advance in the field of regeneration.

The one and important point that was not sufficiently addressed is whether the effects on iNOS are transient. This important to consider for both mechanistic and translational purposes. The first round of reviews raised this point: "Does the tail regenerate when iNos inhibitors are removed?", and the response to this point was a consideration of iNOS regeneration, but not the impact on regeneration if iNOS is again generated after the removal of the inhibitor. Answering this question by removal of the

inhibitor and observing regeneration would be straight forward, and also allow for a cause-and-effect experiment to be performed – for instance, if the L-NAME or PTIO are removed, and the fin does grow back, does this correspond to S-nitrosylation of Kdm1a? If not, this result would require a re-thinking of the conclusions of the study.

Minor point:

1. Please pay attention to zebrafish nomenclature - when discussing proteins, the nomenclature guidelines indicate it is non-italics, first letter upper case (see ZFIN).
2. In figure 6A there is shift in the amino acid sequence of Kdm1a WT that needs to be corrected.
3. In Figure 6B, second sample from left, it should be only Kdm1a Mo, and not be with RNA rescue.

We would like to thank the Editor and Reviewers for the constructive remaining comments. We have made the necessary and substantive changes to the manuscript according to the recommendations. We believe that the manuscript has been further improved based on these insightful comments and is now ready for publication in *Nature Communications*.

Reviewer #2 (Remarks to the Author):

While the authors have addressed many of my previous comments and their biochemical analysis is excellent, there are still significant major and minor concerns on zebrafish regeneration experiments.

Majors

1. *kdm1a* expression pattern during regeneration

*The authors demonstrated that *kdm1a* expression is significantly higher in the differentiating region. To do this, the authors used real-time PCR (quantitative RT-PCR) for whole, distal and proximal regenerates. However, there is no detailed method describing how to collect distal and proximal regions. Indeed, collecting distal and proximal regions separately by dissection is very challenging without using microdissection or FACS sorting with transgenic fish.*

We apologise to have missed this information. We have now included in the methods section, page 20-21 lines 489-494, the protocol to collect the different tailfin regions. We performed these experiments using *Tg(fli1:EGFP)* zebrafish, where endothelial cells fluoresce green. Under a fluorescence stereo microscopes Leica M205 we microdissected at 3 and 5 day post-amputation the regenerated regions, that clearly showed the formation of newly forming (GFP+) vessel branches (see Fig. 5A), and the distal regions, that does not show GFP+ cells. We used the distal ends of the newly forming vessel branches to separate the two regions. We placed the tissues in Eppendorf tubes and immediately extracted RNA that then was used to perform reverse transcription and PCR.

*The authors should examine where *kdm1a* expresses by other methods. The potential method is in situ hybridization. *kdm1a* probe was described in the previous paper (Takeuchi et al., PNAS, 2015). Another possible approach is to analyze single-cell data. Several single-cell profiles are available: Hou et al., Science Adv. 2020; Wang et al., Science 2020; Alemany et al., 2018 Nature. Another approach is to do qPCR with positive control genes, such as blastema or cell cycle genes and differentiation marker genes, to validate distal and proximal regenerate samples.*

Thank you for this insightful comment. As KDM1a has been implicated in several physiological and physiopathological processes occurring in different cell types, we were interested to analyse how KDM1a was modulated also at single cell level during the tailfin regeneration. We looked at the single-cell RNA-seq dataset obtained from zebrafish regenerating tailfin published by Hou et al. (*Science Advances*, 2020), as suggested by this reviewer. Our bioinformatic analysis of this dataset (method described at page 25, lines 605-614) showed that the expression of *Kdm1a* was low but detectable in several cell types (**Fig. S7B**). Although these datasets cannot be directly compared with our KDM1a expression data, obtained by real-time PCR and at different time-points, this clearly show that KDM1a could be differentially modulated in each cell type and is dynamically regulated. These results have been included at page 9, lines 205-213.

2. Morpholino effect

a) Morpholino validation

Two morpholinos (MOs) were used for knock-down experiments. Although the authors described the effect of MO in Lines 265 – 270, the actual results were not

provided. Moreover, it is unclear whether Fig. 6B was obtained by a single MO injection or two MOs injection.

*The authors mentioned “additive effects”, but it is unclear how to determine the MO effect. The appropriate way to determine the MO effect would be 1) western blot and 2) checking phenotype of *kdm1a* knockdown rather than abnormal development (see below). The authors should provide comprehensive results to inform how the individual or combined MO effect was determined.*

Thank you for this comment. We realise that the explanation of the knockdown strategy in the main text was incomplete, therefore we have now explained it more comprehensively in the text. To confirm the morpholino effects and to avoid sequence-specific off-target effects following injection with the first morpholino, we used a second morpholino which sequence did not overlap the first one with the aim to observe in a separate set of experiments a comparable phenotype. This strategy is commonly used to confirm the specificity of the morpholino for the gene of interest (Stainier *et al.*, Plos Genetic 2017). Indeed, both separated morpholino injections were able to reduce KDM1a expression and produced similar phenotype(s). To further confirm the specificity of these morpholinos, as also suggested in Bill *et al.* (Zebrafish, 2009), in a separate set of experiments we co-injected half-doses of both oligos, such that the phenotype was only apparent with each oligo alone, that produced additive effects on the phenotype. This additive effect is a strong indication that the MO effect is specific to the targeted gene. We have now added this explanation at page 12 lines 286-296.

In conclusion, two non-overlapping morpholinos were used to assess the specificity of the oligos for KDM1a. Once the specificity was confirmed, one morpholino was used for further experiments, which results are shown in figures 6 and S9.

*The previous study (Takeuchi et al., PNAS, 2015) demonstrated that *kdm1a* mutation impairs hemangioblast development, leading to less blood cell formation and reduced *gata1* expression. *kdm1a* mutants unlikely exhibit abnormal appearances, such as edema or curved body (Fig. S2B in PNAS paper). The majority of *kdm1a* mutants can survive until 8 dpf in the reference. In the current manuscript, the authors validated morpholino (MO) effects by assessing the survival rate, appearance, *gata1* gene expression and measuring the blood velocity. First, Fig. S9 is not appropriate since it is difficult to know the phenotype of each group. The authors should change the format based on each treated sample rather than phenotypes. It is likely that regardless of MO type MO injection caused severe developmental defects, which raised questions about integrity. The authors should clarify whether normal embryos were used for further analysis or not. Second, the previous report did not test the blood velocity of *kdm1a* mutants. Please provide the rationale for why the authors measure the blood velocity. It is unclear why this analysis can represent *kdm1a* function. The more important phenotype of *kdm1a* should be hematopoietic defects and upregulation of endothelial markers (preferably tested by *in situ* hybridization).*

There are indeed differences between our data on the KDM1a and those included in the report from Takeuchi *et al.* (PNAS, 2015). However, these discrepancies depend on the different embryo developmental stages and groups (morphant or mutant) analysed. For example, in Takeuchi *et al.*, the KDM1a morpholino data, reported in fig. S1, were obtained in embryos at 14 hours post-fertilisation, showing *gata1* reduction in KDM1-Mo injected embryos. Our study not only confirmed the reduced *gata1* expression following KDM1a KD, but also reports, at later developmental stages, specifically at 72 hpf (**Fig. S9**), additional details including abnormal phenotypic features, survival curve and blood velocity, not

reported in Takeuchi et al. Furthermore, the data in Fig. S2B in the PNAS paper are obtained in KDM1a mutants and not in morpholino-injected larvae. Indeed, Takeuchi et al., showed that the effects of the maternal KDM1a, either as protein or mRNA, could explain the lower mortality of zebrafish mutants compared to KDM1a KO mice, which die around 6.5–7.5 d post-coitum. As morpholinos are injected at 1-2 cell stages, they could impair the translation of maternal mRNA as well, therefore causing additional effects on the larval development.

Following the suggestion from this reviewer, we have accordingly changed the format of Fig. S9D that is now based on each treated sample rather than phenotypes. We have now included the exact number, not the percentage, of larvae with a given phenotypic feature out of the total of n=200 embryos per group injected with morpholino. Embryos showing strong developmental defects were not selected for further experiments. In conclusions, we believe that our KDM1a data and those reported in the PNAS paper are complementary as they have used different methods of analysis.

The rationale to analyse blood flow, or blood velocity, is that this is a measure of the cardiovascular function (Yalcin H et al. *Developmental Dynamics.*, 2017) and show a functional feature following KDM1a KD. Our data showed a reduced blood velocity following KDM1a KD and are supported by Takeuchi et al. In fact, their movies S03 and S04 not only show a reduced number of blood cells associated to defects in the differentiation of erythroid cells, but also a clear reduction of blood velocity in *Lsd1* mutants, that unfortunately they did not further describe in the manuscript.

b) kdm1a roles for the regulation of vasculogenesis gene expression

The authors treated PTIO to examine the effect of KDM1a. However, PITO treatment (Fig. 5) does not directly target KDM1a. The authors should examine whether kdm1a KD causes reduced expression of vasculogenesis genes and reduced enrichment of H3K4me2 (ChIP-PCR). Also, the author should provide a detailed method where they target for ChIP experiment. How to validate promoter, the length of PCR region, etc. The authors used the FINDM software to identify promoters, but it is unclear what the FINDM software is.

The reason why we treated zebrafish with PTIO (**Fig. 5**) was to examine the effect of this nitric oxide scavenger on the S-nitrosylation (S-NO) of KDM1a, and not on the whole protein, during tailfin regeneration. Indeed, PTIO inhibited the S-NO of KDM1a but did not affect the expression of KDM1a, as shown in **Fig. 5E**, and this was sufficient to reduce the enrichment of H3K4me2 on *vegfaa* and *tek* genes (**Fig. 5G**), favour their expression (**Fig. 5H**) and ultimately reduce vessel density in the regenerating tissue (**Fig. 5F**). We understand that the PTIO can affect the S-NO of other proteins as well, but unfortunately a specific inhibitor of the S-NO of KDM1a is not available yet. Examining whether *Kdm1a* KD reduces the expression of vasculogenesis genes and or reduces the enrichment of H3K4me2 on these genes is out of the scope of this work, which principal aim is to analyse the role of the S-nitrosylation.

We have now included in the methods section more details on how we performed the ChIP-PCR (see methods section, page 26 lines 624-633). To identify the gene promoter, first we looked in ENSEMBL the gene sequence up to 600bp upstream of the TSS. Then, we validated this sequence on genome.ucsc.edu to confirm it was upstream of our gene of interest. We also looked for the presence of CpG islands and TATA box. Hence, we designed four couple of primers for each gene that match in this region and around the TSS and that could produce amplicons of size no more than 120-130 bp so that both primers could find their targets on one fragment of ChIP DNA, if present. Primer Blast was used to generate

primers. In-silico PCR (UCSC) was used to confirm that primers match our region of interest and the amplicon size, and then primers were further validated by PCR using genomic DNA.

c) Blood vessel density

There is no detailed method for blood vessel density analysis. The authors should also provide representative images used for Fig. 5F.

We have now included the method we used to analyse the vessel density, page 21 lines 495-502.

d) Systemic effects

Please discuss the potential interpretation of systemic effects of *kdm1a* through immune cell regulation on fin regeneration in the discussion section.

We have now discussed the potential roles of the systemic effects of *kdm1a* modulation through immune cell regulation on fin regeneration, page 13-14 lines 317-324.

3. S-nitrosylated nuclear proteins

The authors need to provide a more in-depth analysis of LC/MS/MS to identify S-nitrosylated nuclear proteins. The majority of proteins identified by LC/MS/MS should be nuclear proteins, but it is not demonstrated. Please describe how many percentages of identified proteins are nuclear proteins. Also, please revise dataset S1 to provide more information, such as gene names, a brief description of the function, and source (similar to Table S1).

To effectively separate nuclear proteins from the cytosolic proteins, we used the NE-PER extraction reagents kit (see method section pag. 21 lines 503-519), already used in previous studies (<https://www.thermofisher.com/order/catalog/product/78833?SID=srch-srp-78833#/78833?SID=srch-srp-78833>). In the western blotting in Fig. S1D, we demonstrated the effective separation of nuclear and cytosolic protein fractions using β -tubulin and Histone H3, broadly used as controls respectively for cytosolic and nuclear proteins (for example as in West KO *et al.*, Cell Reports, 2020). Nonetheless, it is very difficult to state confidently whether or not a protein is specifically cytosolic or nuclear, in particular if chemically modified by S-nitrosylation, a posttranslational modification that has been shown to affect the subcellular localisation of proteins (Kornberg *et al.*, Nat cell Biol, 2010).

We have now revised the dataset S1, as suggested by this reviewer, and added the full gene name and source.

4. Drug delivery method

First, I do recommend remove the first sentence in “the optimization of an injection procedure”: “we know that even 1 uL of physiological solution injected daily for a week can easily cause death, maybe because of the damage that the needle, even when used by expert hands, can cause in such small animals.” This sentence may mislead researchers to consider that injecting a drug in adult fish is challenging.

We have now removed this sentence.

However, many zebrafish labs using adults routinely inject drugs without killing fish. The major reasons to kill fish by injection are likely 1) high dose of drug 2) high percentage of solvent, and 3) an injection tool. For DMSO and Ethanol, the maximum percentage would be 10-20%. The authors did not describe how they prepare for drugs (conc. of stock, solvent to dissolve drug, etc). The authors seem to use a glass capillary but do not describe how to prepare it. The common injection tool for fish is a Hamilton syringe (1700 series having a customized gauge), not a

glass capillary. Note that the cited reference 34 also used a Hamilton syringe. A glass capillary diameter may be large to significantly damage fish, leading to death. Although functional data likely support the effect of drug treatment, the authors need to clarify that doses used for regeneration study do not significantly affect mortality, like LNAME.

Thank you for this comment. We have now improved our description of drug preparations, including solvent used to dissolve, stock concentration, etc., and described it in “Pharmacological modulation of S-nitrosylation” in the methods section.

As regarding the injection procedure, we normally use glass capillaries because we feel they offer more flexibility as we can better customise the tip diameter. Indeed, in Pugack EK *et al.*, J Vis Exp 2009 (ref. 34 in the previous version, ref. 40 in the present version) they do not specify the size of the needle mounted on their Hamilton syringe. Nonetheless, by using a micropipette puller (Narishige, PC-10) and selecting optimal parameters, we can obtain glass micropipette with an outer diameter at the tip of up to 60 microns (Samaee S. *et al.*, Zebrafish 2017) that is lower of the outer diameter (OD) of a 34-gauge (the tiniest OD available) needle mounted on a Hamilton syringe, including the 1700 series. According to our experience, this glass micropipettes help us to reduce the damage to the fish eye.

The dose of drugs, like L-Name, used in our experiments on tailfin regeneration did not affect the mortality compared to control. To show this, we have now included the statistical analysis in **Fig. S2B**, using the Log-rank test and the Gehan-Breslow-Wilcoxon Test. The test used were reported in the respective figure legend and in “Statistical analysis” in methods section.

Minor

1. Nomenclature: Please check the correct nomenclature for zebrafish genes, proteins, transgenic lines throughout the manuscript.

Thank you to point this out. We have now made the necessary amendments, using the correct nomenclature adopted for zebrafish.

2. “Baseline” in multiple figures -> please change to uninjured throughout figures and manuscript

We have now fixed this.

3. RT-qPCR

Fig 1a, S5, S7, S9. Y-axis is weird. Please revise the number. There is no information about primers. If Table S2 contains primers for qRT-PCR, please clarify it. Also, there is no primer information for b-actin.

We have now revised the numbers on Y-axes as requested. In particular, in Fig 1a, S5 and S7 we have adopted the scientific format, whereas in Fig. S9 we have reduced the decimal places at two (in place of three). We have also mentioned the list of primers included in **Table S2** and used to perform PCR in the method section, line 766. We have now included in Table S2 also the primers used to assess b-actin.

4. Timepoint to assess regeneration

Please clarify the time point (regeneration or injured) and/or drug concentration in Fig. 1F, 3D, 3E, 3F, 4D, 4E, 5B-G, 6BD and many supplementary figures.

5. Fig 3. Please add “IP:KDM1a” for E and F above the figures.

We have now fixed this. The missing information has been added in the figures and respective legends.

Reviewer #3 (Remarks to the Author):

In this revised version of the manuscript, the authors completed several new experiments that further support the conclusions that S-nitrosylation of KDM1a impairs its ability to demethylated H3K4 and this impairs fin regeneration in zebrafish. The model is elegant and several new insights are provided by the work. However, there remain a few outstanding issues that were not addressed. It would be beneficial for the authors to state outright if they are unable to address these points experimentally or do the experiments which address them directly. The new experiments showing that endothelial cells are a primary target of iNOS inhibition and KDM1a action are very interesting. Overall, the manuscript is improved and makes an interesting advance in the field of regeneration.

Thank you for the kind comment.

The one and important point that was not sufficiently addressed is whether the effects on iNOS are transient. This important to consider for both mechanistic and translational purposes. The first round of reviews raised this point:

“Does the tail regenerate when iNos inhibitors are removed?”, and the response to this point was a consideration of iNOS regeneration, but not the impact on regeneration if iNOS is again generated after the removal of the inhibitor. Answering this question by removal of the inhibitor and observing regeneration would be straight forward, and also allow for a cause-and-effect experiment to be performed – for instance, if the L-NAME or PTIO are removed, and the fin does grow back, does this correspond to S-nitrosylation of Kdm1a? If not, this result would require a re-thinking of the conclusions of the study.

Apologies that our response in the previous revision was not exhaustive. The answer to the question “Does the tail regenerate when iNos inhibitors are removed?” is yes. To further clarify this, we have now included in **Fig. S2A** the analysis of tailfin regeneration at 3 days post-amputation (dpa) after the suspension of L-NAME injections in the zebrafish (last injection was performed at the day of the amputation) and compared this with tailfin regeneration at 7 dpa (**Fig. 1F**). **Fig S2A** shows that the percentage of tailfin regeneration at 3 dpa was 20.2 ± 1.2 in control (an average of $\sim 6.7\%$ per day) vs 8.9 ± 0.9 (an average of $\sim 3\%$ per day) in L-NAME treated group, whereas at 7 dpa (**Fig. 1F**) it was 59.5 ± 1.7 in control (a 39.3% increase from day 4 to day 7 post-amputation, that is an average of $\sim 9.8\%$ of regeneration per day) vs 44.6 ± 1.3 in L-NAME treated group (a 35.7% increase from day 4 to day 7 post-amputation, an average of $\sim 8.9\%$ of regeneration per day). Therefore, while in the first 3 dpa the percentage of regeneration was more than 100% higher in control vs L-NAME treated group, at 7 dpa this percentage was only 33.5 % higher in control vs treated group. This clearly showed that at 7 dpa, when the inhibitory effect of L-NAME on iNOS loosened, the tailfin regeneration started to regenerate at a pace similar as in control. A similar trend in tailfin regeneration was observed following treatment of zebrafish with the NO scavenger PTIO (**Fig. S2A**). To demonstrate the causal nexus between PTIO and S-nitrosylation (S-NO) of KDM1a, in **Fig. 5E** we included data showing that the treatment with PTIO almost abolished the S-NO of KDM1a, specifically in endothelial cells of the regenerating tailfin, and this was sufficient to reduce the enrichment of H3K4me2 on *vegfaa* and *tek* genes (**Fig. 5G**) and the expression of these and other proangiogenic genes (**Fig. 5H**).

Minor point:

1. ***Please pay attention to zebrafish nomenclature - when discussing proteins, the nomenclature guidelines indicate it is non-italics, first letter upper case (see ZFIN).***

Thank you to point this out. We have now used the correct the nomenclature.

2. ***In figure 6A there is shift in the amino acid sequence of Kdm1a WT that needs to be corrected.***

We have now fixed this.

3. ***In Figure 6B, second sample from left, it should be only Kdm1a Mo, and not be with RNA rescue.***

We have now fixed this.

REVIEWERS' COMMENTS

Reviewer #2 (Remarks to the Author):

I appreciate the authors for addressing comments to improve their manuscript. I have some minor comments.

Microdissection

I do not believe that the authors can say “microdissection” for the samples collected using the stereomicroscope. It is unclear how high magnification the authors used, but dissection was likely done by hand. To me, “Microdissection” would be okay to use in case the authors use the specialized microscope, such as Laser Microdissection LMD or LCM (<https://www.leica-microsystems.com/solutions/life-science/laser-microdissection/>). I do not believe that dissection by hand can be called as “microdissection”. Please switch “microdissection” with “dissection”. This term may mislead the readers.

Nomenclature

The authors likely use the right form in the main text and figure legend, but not in figures.

Please revise figures to use the right one.

Fig. 1A uses RT-qPCR, the gene names should be italic lowercase.

Nos1, Nos2a, Nos2b -> *nos1*, *nos2a*, *nos2b* (all italic)

As Fig. 1C graph indicates Nos2 protein and the author used Nos2 as a protein in the main text, NOS2 should be changed to *Nos2*.

Fig. 5F-H. All genes and transcripts, but not proteins, should be lowercase italic.

Fig. 6A. *Kdm1a* -> *kdm1a* (italic)

Fig. S1C, not *Nos2* -> *nos2* (italic) as it indicates gene

Fig. S5: as RT-qPCR, so the authors indicate genes, not proteins. please use *rcor1*, *rbbp*, *hdac1*, *chdrb*, *ar* (all italics).

Fig. S7. *Kdm1a* -> *kdm1a* (italic)

Line 206, *Kdm1a* -> *kdm1a* (italic)

Line 208 typo *KDM1a*

Fig. S8. All genes should be italic lowercase

Fig. S9. All genes should be italic lowercase

Table S2 may indicate gene. If so, please use italic lower case.

Others: line 678, 690, 695 and so on in Methods.

As I may miss some, please check all parts one more time.

MO experiments

If inject two MOs, please use plural form: MOs, not MO. For example, Fig. 6B.

I am still confused with Fig. S9D.

1) Fig. S9D. unlikely contains normal phenotype. Please check one more time whether this bar graph includes all phenotypes and all tested number of embryos. The authors described to test 200 embryos and the total number of control is not 200.

2) It is unclear what “chorionated” indicated. If this indicate unhatched embryos, then what is the result after removing chorion.

3) Some larvae may show two phenotypes, such as reduced swim and reduced length. Does this mean that the embryo exhibiting reduced swim does NOT show any other phenotypes?

4) In fact, I suggest to divide this bar graph into two.

First one indicates embryos showing strong developmental defects. The authors may combine any abnormal embryos as one category rather than sub-divided by curved body, chorionated, edema, and so on. This graph may demonstrate how many embryos display strong developmental defects so that it will tell embryo number used for the next experiments, measuring fin length. Second one is for only embryos used for measuring lengths. The current bar graph unlikely supports *kdm1a* MOs phenotypes is rescued by mRNA injection, which is not true.

Reference

Please check the reference as there is no 45 (used in the introduction).

Dataset S1 and Table S1

It is great to provide this table, but still difficult to know which proteins/peptides are upregulated in given time points. Would it be possible to provide a table(s) indicating what are enriched proteins/peptides in each time points (related to Fig. 2C)? For example, with a table, the readers may recognize what 17 genes are enriched in all 4 times points.

Reviewer #3 (Remarks to the Author):

The authors have sufficiently addressed all the issues raised. The one change I recommend that would improve the manuscript is to break the introduction into paragraphs that focus on the topics under consideration. Also, there is some issue with the numbering of the references that needs to be corrected.

Many thanks to the Reviewers for the additional comments that have allowed us to further improve the manuscript.

Response to Reviewers' comments

Reviewer #2 (Remarks to the Author):

I appreciate the authors for addressing comments to improve their manuscript. I have some minor comments.

Microdissection

I do not believe that the authors can say “microdissection” for the samples collected using the stereomicroscope. It is unclear how high magnification the authors used, but dissection was likely done by hand. To me, “Microdissection” would be okay to use in case the authors use the specialized microscope, such as Laser Microdissection LMD or LCM (<https://www.leica-microsystems.com/solutions/life-science/laser-microdissection/>). I do not believe that dissection by hand can be called as “microdissection”. Please switch “microdissection” with “dissection”. This term may mislead the readers.

Answer: We have now changed “microdissection” with “dissection”.

Nomenclature

The authors likely use the right form in the main text and figure legend, but not in figures. Please revise figures to use the right one.

Answer: We have now amended nomenclature in the figures.

Fig. 1A uses RT-qPCR, the gene names should be italic lowercase.

Nos1, Nos2a, Nos2b -> nos1, nos2a, nos2b (all italic)

As Fig. 1C graph indicates Nos2 protein and the author used Nos2 as a protein in the main text, NOS2 should be changed to Nos2.

Fig. 5F-H. All genes and transcripts, but not proteins, should be lowercase italic.

Fig. 6A. Kdm1a -> kdm1a (italic)

Fig. S1C, not Nos2 -> nos2 (italic) as it indicates gene

Fig. S5: as RT-qPCR, so the authors indicate genes, not proteins. please use rcor1, rbbp, hdac1, chdrb, ar (all italics).

Fig. S7. Kdm1a -> kdm1a (italic)

Line 206, Kdm1a -> kdm1a (italic)

Line 208 typo KDM1a

Fig. S8. All genes should be italic lowercase

Fig. S9. All genes should be italic lowercase

Table S2 may indicate gene. If so, please use italic lower case.

Others: line 678, 690, 695 and so on in Methods.

As I may miss some, please check all parts one more time.

Answer: We have now corrected these mistakes.

MO experiments

If inject two MOs, please use plural form: MOs, not MO. For example, Fig. 6B.

Answer: Fig. 6B data were obtained by injecting one morpholino.

I am still confused with Fig. S9D.

1) Fig. S9D. unlikely contains normal phenotype. Please check one more time whether this bar graph includes all phenotypes and all tested number of embryos. The authors described to test 200 embryos and the total number of control is not 200.

Answer: We have now included also the number of normal embryos in each group.

2) It is unclear what “chorionated” indicated. If this indicate unhatched embryos, then what is the result after removing chorion.

Answer: Zebrafish embryos dechorionate at approximately 72 hours post fertilization (3 days post fertilization), and sporadically during the whole third day of development (at 28 °C). Embryos still “chorionated” at later stages could be associated with developmental delay (Jeanray N et al., Phenotype Classification of Zebrafish Embryos, PLOS one 2015), or with abnormal phenotypes such as curved body, reduced length, etc. Therefore, we considered embryos chorionated at 4 dpf as abnormal. These embryos were not used for further experiments.

3) Some larvae may show two phenotypes, such as reduced swim and reduced length. Does this mean that the embryo exhibiting reduced swim does NOT show any other phenotypes?

4) In fact, I suggest to divide this bar graph into two.

First one indicates embryos showing strong developmental defects. The authors may combine any abnormal embryos as one category rather than sub-divided by curved body, chorionated, edema, and so on. This graph may demonstrate how many embryos display strong developmental defects so that it will tell embryo number used for the next experiments, measuring fin length. Second one is for only embryos used for measuring lengths. The current bar graph unlikely supports kdm1a MOs phenotypes is rescued by mRNA injection, which is not true.

Answer: we understand that the description of the phenotype in Fig.S9D can generate confusion, as several embryos show more abnormal features at the same time.

Therefore, phenotype data for each embryo group have now been reported graphically as “normal”, i.e. embryos not showing any abnormal features, and “abnormal”, i.e. embryos showing one or more of the features described above. This has also been described in the legend of fig. S9 and in the Methods section, lines 645-646. Embryos with abnormal features were not used for further experiments.

Reference

Please check the reference as there is no 45 (used in the introduction).

Answer: Those are references 4 and 5 of Introduction. Sorry for the mistake. We have now added a comma between 4 and 5 to avoid misunderstanding.

Dataset S1 and Table S1

It is great to provide this table, but still difficult to know which proteins/peptides are upregulated in given time points. Would it be possible to provide a table(s) indicating what are enriched proteins/peptides in each time points (related to Fig. 2C)? For example, with a table, the readers may recognize what 17 genes are enriched in all 4 times points.

Answer: Yes, we have now included an additional sheet in the dataset S1, named Venn Diagram, with the information on enriched proteins and peptides reported in the Venn diagram.

Reviewer #3 (Remarks to the Author):

The authors have sufficiently addressed all the issues raised. The one change I recommend that would improve the manuscript is to break the introduction into paragraphs that focus on the topics under consideration.

Answer: We have now divided the Introduction into three paragraphs.

Also, there is some issue with the numbering of the references that needs to be corrected.

Answer: We have made these corrections.